# Disruption of the novel nested gene *Aff3ir* mediates disturbed flow-induced atherosclerosis in mice

Shuo He[1†], Lei Huang[2†], Zhuozheng Chen[1], Ze Yuan[1], Yue Zhao[2,3], Lingfang Zeng[3*], Yi Zhu[1*], Jinlong He[1*]

[1]Province and Ministry Co-sponsored Collaborative Innovation Center for Medical Epigenetics; NHC Key Laboratory of Hormones and Development; Department of Physiology and Pathophysiology, Tianjin Medical University, Tianjin, China; [2]Department of Heart Center, The Third Central Hospital of Tianjin; Tianjin Universiy Central Hospital; Tianjin Key Laboratory of Extracorporeal Life Support for Critical Diseases; Artificial Cell Engineering Technology Research Center; Tianjin Institute of Hepatobiliary Disease; Nankai University Affinity the Third Central Hospital, Tianjin, China; [3]School of Cardiovascular and Metabolic Medicine and Sciences, King's College London British Heart Foundation Centre of Excellence, Faculty of Life Sciences and Medicine, King's College London, London, United Kingdom

*For correspondence:
lingfang.zeng@kcl.ac.uk (LZ);
zhuyi@tmu.edu.cn (YZ);
hejinlong@tmu.edu.cn (JH)

[†]These authors contributed equally to this work

## eLife Assessment

The study presents **valuable** findings on the role of Aff3ir, a gene implicated in flow-induced atherosclerosis and regulating the inflammation-associated transcription factor, IRF5. The in vivo data are **solid** in providing evidence on the role of Aff3ir in shear stress and formation of atheromatous plaques. The work will be of interest to clinical researchers and biologists focusing on inflammation and atherosclerosis in cardiovascular disease with a broad eLife readership.

**Abstract** Disturbed shear stress-induced endothelial atherogenic responses are pivotal in the initiation and progression of atherosclerosis, contributing to the uneven distribution of atherosclerotic lesions. This study investigates the role of *Aff3ir-ORF2*, a novel nested gene variant, in disturbed flow-induced endothelial cell activation and atherosclerosis. We demonstrate that disturbed shear stress significantly reduces *Aff3ir-ORF2* expression in athero-prone regions. Using three distinct mouse models with manipulated *Aff3ir-ORF2* expression, we demonstrate that *Aff3ir-ORF2* exerts potent anti-inflammatory and anti-atherosclerotic effects in *Apoe*[-/-] mice. RNA sequencing revealed that interferon regulatory factor 5 (*Irf5*), a key regulator of inflammatory processes, mediates inflammatory responses associated with *Aff3ir-ORF2* deficiency. *Aff3ir-ORF2* interacts with *Irf5*, promoting its retention in the cytoplasm, thereby inhibiting the *Irf5*-dependent inflammatory pathways. Notably, *Irf5* knockdown in *Aff3ir-ORF2* deficient mice almost completely rescues the aggravated atherosclerotic phenotype. Moreover, endothelial-specific *Aff3ir-ORF2* supplementation using the CRISPR/Cas9 system significantly ameliorated endothelial activation and atherosclerosis. These findings elucidate a novel role for *Aff3ir-ORF2* in mitigating endothelial inflammation and atherosclerosis by acting as an inhibitor of *Irf5*, highlighting its potential as a valuable therapeutic approach for treating atherosclerosis.

## Introduction

Atherosclerosis, characterized by the formation of fibrofatty lesions in the arterial wall, is a leading cause of morbidity and mortality worldwide, contributing to most myocardial infarctions and many strokes (*Herrington et al., 2016*; *Libby et al., 2019*). The activation of vascular endothelial cells (ECs), induced by various chemical and mechanical stimuli, such as lipopolysaccharide and shear stress, is an initial step in the development of atherosclerosis (*Davignon and Ganz, 2004*). Consequently, atherosclerotic lesions preferentially develop at the branches and curvatures of the arterial tree, where blood flow is disturbed (*Davis et al., 2023*). For decades, researchers have been interested in exploring the mechanisms underlying mechanotransduction during endothelial activation caused by disturbed flow (*Davis et al., 2023*). Several mechanosensitive proteins, such as Yap/Taz (*Li et al., 2019*), Annexin A2 (*Zhang et al., 2020*), Bmp4 (*Sorescu et al., 2004*), and Nad(p)h oxidase (*Hwang et al., 2003*; *Jo et al., 2006*), have been identified as key regulators of disturbed shear stress in ECs and have been implicated in the progression of atherosclerosis. Emerging evidence has revealed that pharmacological or genetic inhibition of endothelial *Yap* activation ameliorates the progression of atherosclerotic plaques in mice (*Li et al., 2019*; *Wang et al., 2016*; *Yang et al., 2021*), indicating that targeting disturbed flow-induced endothelial activation could be a promising therapeutic strategy for atherosclerosis. However, the precise mechanisms by which the disturbed flow exerts detrimental effects remain unclear.

The interferon regulatory factor (IRF) family of transcription factors, comprising nine members (*Irf1-Irf9*) in mammals, is primarily characterized by its role in mediating antiviral responses and type I interferon production (*Sato et al., 2001*). Although these members share a conserved DNA-binding domain in their N-terminal region that recognizes similar DNA sequences, *Irf5* plays a central role in inflammation (*Almuttaqi and Udalova, 2019*; *Takaoka et al., 2005*). *Irf5* mediates the production of proinflammatory cytokines, including *Il12b* and *Il23a*, and promotes the expression of inflammatory genes (*Cai et al., 2017*; *Saliba et al., 2014*; *Weiss et al., 2013*). It promotes inflammatory responses in various immune cells, including macrophages (*Seneviratne et al., 2017*), neutrophils (*Weiss et al., 2015*), and B cells (*Savitsky et al., 2010*). Global or myeloid-specific knockouts of *Irf5* have been shown to exert anti-atherosclerotic effects (*Leipner et al., 2021*; *Seneviratne et al., 2017*). Despite the established importance of *Irf5* in immune cells, its restrictively regulatory mechanism and role in shear stress-induced endothelial activation remain unknown.

We recently reported that a novel protein-coding nested gene, *Aff3ir*, contributes to endothelial maintenance by promoting the differentiation of vascular stem/progenitor cells (SPCs) into ECs (*Zhao et al., 2025*). *Aff3ir-ORF2*, encoded by the *Aff3ir* transcript variant 2, is predominantly expressed in the EC layer of the mouse aorta (*Zhao et al., 2025*). Notably, our recent study indicated that the overexpression of *Aff3ir-ORF2* could enhance laminar flow-induced mRNA levels of essential EC markers in SPCs (*Zhao et al., 2025*), suggesting that *Aff3ir-ORF2* may be a novel mechanotransduction protein in ECs. However, the regulation of *Aff3ir-ORF2* under disturbed flow and its role in atherosclerosis remain unclear.

In this study, we aimed to elucidate the mechanism by which disturbed blood flow induces endothelial activation and atherosclerosis. Our study showed that disrupted *Aff3ir-ORF2* expression in atheroprone regions led to inflammatory responses and development of atherosclerosis. *Aff3ir-ORF2*, the expression of which is reduced by disturbed shear stress, exerts critical anti-inflammatory effects by binding to *Irf5* and mitigating disturbed shear stress-induced *Irf5* activation. Additionally, we demonstrated that endothelial-specific supplementation with *Aff3ir-ORF2* significantly ameliorated disturbed flow-induced endothelial activation and the development of atherosclerotic plaques, highlighting its promising therapeutic potential for the treatment of atherosclerosis.

## Results

### Disturbed shear stress reduces the expression of *Aff3ir-ORF2*

Our recent study showed the active participation of *Aff3ir* in EC differentiation from vascular SPCs induced by laminar shear stress (*Zhao et al., 2025*), suggesting the potential involvement of this novel protein-encoding nested gene in mediating hemodynamic stimulation. To further elucidate the functional role of *Aff3ir* and its encoded proteins in disturbed shear stress-induced EC activation, we examined the expression of *Aff3* and *Aff3ir* in the intima of mouse aorta. We found that the mRNA

level of *Aff3ir*, but not its parent gene *Aff3*, was significantly lower in the intima of the aortic arch, an area exposed to disturbed shear stress, compared to the intima of the thoracic aorta, which was exposed to steady unidirectional shear stress (*Li et al., 2019*; *Figure 1A*). *Aff3ir* transcript variants can generate two proteins (*Zhao et al., 2025*), therefore, we measured the protein levels of *Aff3ir-ORF1* and *Aff3ir-ORF2*. While *Aff3ir-ORF1* and *Aff3* showed comparable expression levels in the intima of aortic arch and thoracic aorta of mice, the expression of *Aff3ir-ORF2* showed an 87% reduction in the intima of aortic arch compared to the intima of thoracic aorta (*Figure 1B and C*), suggesting that *Aff3ir-ORF2* may be a novel mechanosensitive protein that responds to disturbed shear stress. Enface immunofluorescence staining confirmed a marked reduction in *Aff3ir-ORF2* expression in the inner curvature of aortic arch compared to both the outer curvature of aortic arch and the thoracic aorta (*Figure 1D and E*). Moreover, to demonstrate the change in *Aff3ir-ORF2* within the same visual field, we examined its expression in longitudinal sections of the mouse aorta (*Li et al., 2019*). We found that the expression of *Aff3ir-ORF2*, but not *Aff3*, was notably downregulated in athero-prone regions (the inner curvature of the aortic arch and bifurcation of the carotid artery) compared to that in the protective region in the outer curvature of the aortic arch (*Figure 1F*). Additionally, we found that the expression of *Aff3*, *Aff3ir-ORF1*, and *Aff3ir-ORF2* in the media and adventitia was comparable between the aortic arch and the thoracic aorta (*Figure 1—figure supplement 1A, B*).

Next, we explored the impact of disturbed shear stress on *Aff3ir-ORF2* expression in vitro. Mouse embryonic fibroblasts (MEFs) exhibit responses consistent with those of ECs (*Chen et al., 2010*; *Wen et al., 2013*), therefore, we investigated *Aff3ir-ORF2* expression in MEFs from WT mice exposed to static or disturbed flow ($0.5\pm4$ dyn/cm$^2$, 1 Hz). Consistent with our in vivo findings, while disturbed shear stress increased the expression of *Vcam1*, a critical inflammatory marker of ECs (*Nakashima et al., 1998*), it significantly reduced both the protein and mRNA levels of *Aff3ir-ORF2* (*Figure 1G–I*). The expression of *Aff3* in response to the disturbed flow was minimally affected at both the mRNA and protein levels (*Figure 1G–I*). These results collectively demonstrate that disturbed shear stress induces a reduction in *Aff3ir-ORF2* expression both in vivo and in vitro.

## *Aff3ir-ORF2* ameliorates disturbed shear stress-induced inflammation and atherosclerosis

Disturbed shear stress-induced atherogenic responses are initial events in atherosclerotic plaque formation (*Davis et al., 2023*). To elucidate the regulatory role of *Aff3ir-ORF2* in disturbed shear stress-induced inflammation, we overexpressed *Aff3ir-ORF2* in MEFs. *Aff3ir-ORF2* overexpression attenuated *Icam1* expression induced by disturbed shear stress at both the protein and mRNA levels (*Figure 2A, B*, *Figure 2—figure supplement 1A*). To further validate our findings in ECs, we overexpressed *Aff3ir-ORF2* in human umbilical vein endothelial cells (HUVECs). Consistent with our previous results, *Aff3ir-ORF2* overexpression reduced the protein level of *Icam1* induced by disturbed shear stress in HUVECs (*Figure 2C–D*). Moreover, *Aff3ir-ORF2* overexpression attenuated disturbed shear stress-induced expression of several inflammatory genes, including *Vcam1*, *Il6*, and *Il1b*, in both MEFs and ECs. (*Figure 2—figure supplement 1A–B*). Interestingly, we found that *Aff3ir-ORF2* overexpression did not affect the basal expression of these inflammatory genes under ST conditions (*Figure 2—figure supplement 1A–B*), likely due to the relatively low levels of inflammatory gene expression under ST compared to OSS conditions. Notably, despite the significant anti-inflammatory effects of *Aff3ir-ORF2*, the sequence of this gene is not conserved in *Homo sapiens*. Furthermore, we measured the concentrations of inflammatory factors, including *Il6* and *Il1b*, in the culture medium of MEFs. As expected, while *Aff3ir-ORF2* overexpression had little effect on the concentrations of *Il6* and *Il1b* under ST condition, it significantly reduced their release induced by disturbed shear stress (*Figure 2E*).

Given the anti-inflammatory effects of *Aff3ir-ORF2*, we speculated that it may ameliorate disturbed shear stress-induced inflammation and atherosclerosis in vivo. Apoe knockout (*Apoe$^{-/-}$*) mice were subjected to partial ligation surgery to induce disturbed flow in the left carotid arteries (LCAs). The endothelium of the LCAs was intravascularly infected with adenovirus (Ad-Scramble or Ad-*Aff3ir-ORF2*) prior to surgery (*Nam et al., 2009*; *Zhang et al., 2020*). Enface immunofluorescence staining confirmed the successful *Aff3ir-ORF2* overexpression in the left carotid artery (*Figure 2—figure supplement 1C–D*). Mice infected with Ad-*Aff3ir-ORF2* exhibited a significant decrease in lesion area in the LCAs compared to those infected with Ad-Scramble ($23 \pm 17\%$ vs $63 \pm 14\%$) (*Figure 2F–H*), with no obvious plaque formation observed in the right carotid arteries (*Figure 2—figure supplement 1E*).

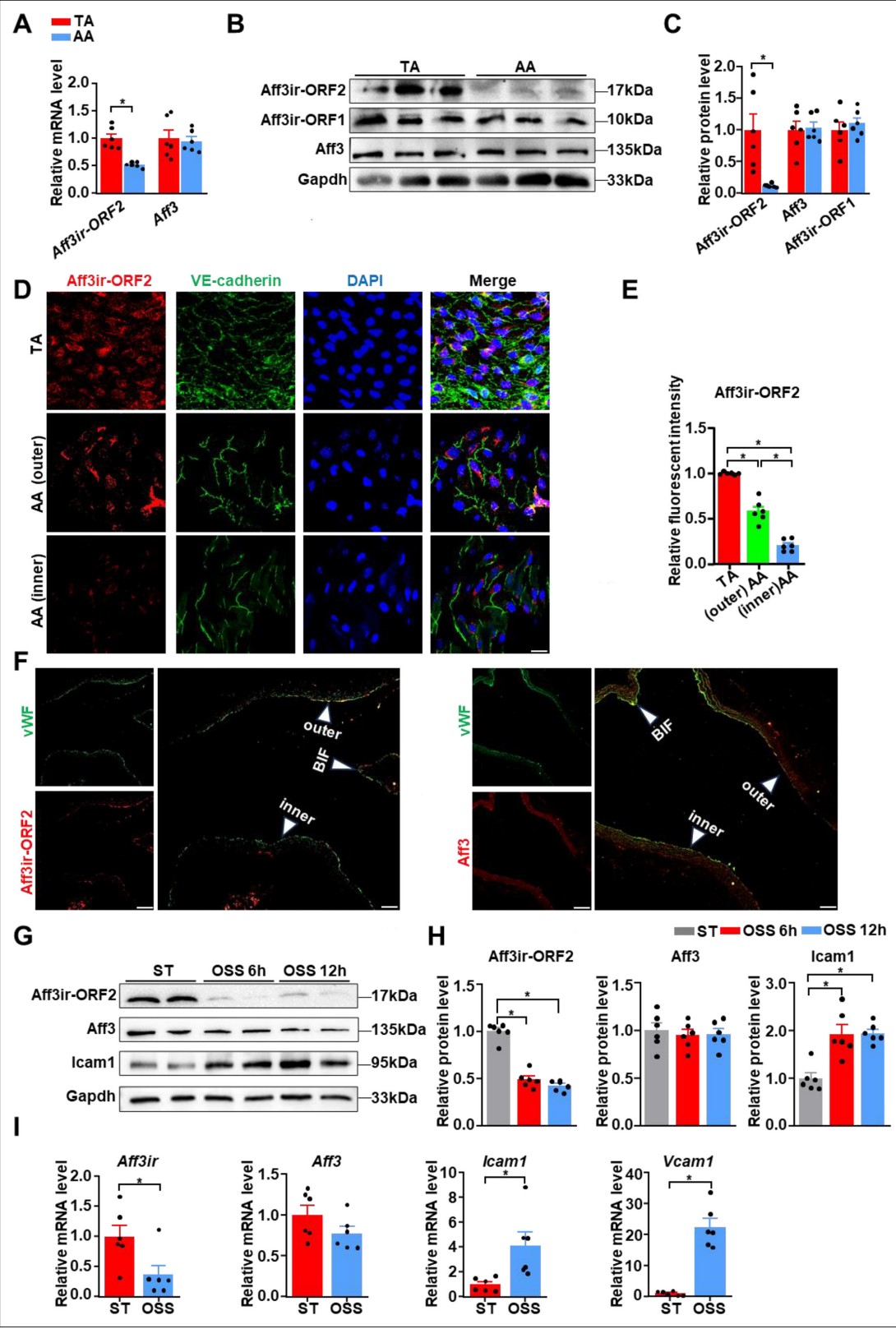

**Figure 1.** Disturbed shear stress reduces the expression of Aff3ir-ORF2 in vivo and in vitro. (**A**) RT-PCR analysis of the mRNA levels of *Aff3ir-ORF2* and AF4/FMR2 family member 3 (*Aff3*) in the intima of thoracic aorta (TA) and aortic arch (AA) of C57BL/6 mice. Data are presented as mean ± SEM (n=6 mice per group). *p<0.05, unpaired two-tailed *t*-test. (**B, C**) Western blot analysis of the expression of the indicated proteins in the intima of TA and AA of C57BL/6 mice. Protein levels were normalized to those of *Gapdh*, and the relative expression values were compared to those of the TA

*Figure 1 continued on next page*

*Figure 1 continued*

group. Data are presented as mean ± SEM (n=6 mice per group). *p<0.05, unpaired two-tailed *t*-test. (**D, E**) En-face immunofluorescence staining of *Aff3ir-ORF2*, VE-cadherin, and DAPI, and quantification of *Aff3ir-ORF2* expression in inner curvature of the AA (AA inner), outer curvature of the AA (AA outer), and TA of C57BL/6 mice. Scale bar, 20 μm. The immunofluorescence intensity of *Aff3ir-ORF2* was normalized to that of DAPI, and the relative expression values were compared to that of the TA group. Data are presented as mean ± SEM (n=6 mice per group). *p<0.05, one-way ANOVA with Tukey post-test. (**F**) Representative immunofluorescent staining for von Willebrand factor (vWF), *Aff3ir-ORF2*, and *Aff3* in longitudinal aortic sections of C57BL/6 mice. n=6 mice per group. Scale bar, 25 μm. Inner, inner curvature of the AA; outer, outer curvature of the AA; BIF, Bifurcation. (**G, H**) Mouse embryonic fibroblasts (MEFs) isolated from the embryo of C57BL/6 mice were subjected to static (ST) or oscillatory shear stress (OSS, 0.5±4 dyn/cm$^2$, 1 Hz) for indicated time. Western blot analysis of the indicated proteins. Protein levels were normalized to *Gapdh* and the relative expression values were compared to that of the ST group. Data are mean ± SEM (n=6 independent experiments). *p<0.05, one-way ANOVA with Tukey post-test. (**I**) MEFs were subjected to ST or oscillatory shear stress (OSS) treatment for 6 hr. RT-PCR analysis of the mRNA levels of *Aff3ir*, *Aff3*, intercellular adhesion molecule 1 (*Icam1*), and vascular cell adhesion molecule 1 (*Vcam1*) in MEFs. Data are mean ± SEM (n=6 independent experiments). *p<0.05, unpaired two-tailed *t*-test.

The online version of this article includes the following source data and figure supplement(s) for figure 1:

**Source data 1.** The table summarizes the data in the statistical graph for *Figure 1A, C, E, H and I*.

**Source data 2.** The original file of the full raw uncropped, unedited polyacrylamide gels for *Figure 1B*.

**Source data 3.** Figures with the uncropped polyacrylamide gels with the relevant bands clearly labeled for *Figure 1B*.

**Source data 4.** The original file of the full raw uncropped, unedited polyacrylamide gels for *Figure 1G*.

**Source data 5.** Figures with the uncropped polyacrylamide gels with the relevant bands clearly labeled for *Figure 1G*.

**Figure supplement 1.** Disturbed shear stress does not reduce the expression of A3ir-ORF2 in the media and adventitia.

**Figure supplement 1—source data 1.** The table summarizes the data in the statistical graph for *Figure 1—figure supplement 1B*.

**Figure supplement 1—source data 2.** The original file of the full raw uncropped, unedited polyacrylamide gels for *Figure 1—figure supplement 1A*.

**Figure supplement 1—source data 3.** Figures with the uncropped polyacrylamide gels with the relevant bands clearly labeled for *Figure 1—figure supplement 1A*.

Overexpression of *Aff3ir-ORF2* also attenuated disturbed flow-induced inflammatory responses, as evidenced by decreased *Vcam1* expression in the endothelium of LCAs (*Figure 2I–J*). These findings suggested that *Aff3ir-ORF2* ameliorates shear stress-induced inflammation and atherosclerosis.

## *Aff3ir-ORF2* deficiency aggravates inflammation and atherosclerotic lesions in *Apoe*[-/-] mice

To explore the effects of *Aff3ir-ORF2* on inflammation and atherosclerosis, we generated *Aff3ir-ORF2* global knockout (*Aff3ir-ORF2*[-/-]) mice. Genotyping PCR (*Figure 3—figure supplement 1A*) and western blot analysis of *Aff3ir-ORF2* expression in mouse aortas (*Figure 3—figure supplement 1B–C*) confirmed the successful knockout. No obvious phenotypic abnormalities were observed in *Aff3ir-ORF2*[-/-] mice up to 20 wk of age and monitoring was discontinued thereafter. Additionally, *Aff3ir-ORF2* deficiency did not alter systolic blood pressure, diastolic blood pressure, or mean arterial pressure (*Figure 3—figure supplement 1D*), suggesting that *Aff3ir-ORF2* is dispensable for physiological blood pressure maintenance. We then isolated MEFs from WT and *Aff3ir-ORF2*[-/-] mice. RT-PCR analysis confirmed the deficiency of *Aff3ir-ORF2* in *Aff3ir-ORF2*[-/-] MEFs (*Figure 3—figure supplement 1E*). Interestingly, the *Aff3ir-ORF2* knockdown efficiency showed discrepancies between the western blot (*Figure 3—figure supplement 1B*) and RT-PCR results (*Figure 3—figure supplement 1E*). In addition to the technical differences between PCR and western blot, the characteristics of *Aff3ir-ORF2* may also contribute to this inconsistency. The parent gene, *Aff3*, is located in a genetically variable region, and it can be excised via intron 5 to form a replicable transposon that translocates to other chromosomes, potentially contributing to leukemia (*Chinen et al., 2008*; *Hiwatari et al., 2003*; *Miller et al., 2022*; *Bergh et al., 2002*). *Aff3ir*, located in intron 6, exists within this transposon, which may complicate the measurement of its expression. Furthermore, we found that *Aff3ir-ORF2* deficient MEFs displayed higher expression of inflammatory genes, including *Icam1*, *Vcam1*, and *Il1b*, compared to those in WT MEFs, under disturbed flow stimulation (*Figure 3—figure supplement 1F*).

Next, we crossed *Aff3ir-ORF2*[-/-] mice with *Apoe*[-/-] mice to generate double-knockout (*Apoe*[-/-]*Aff3ir-ORF2*[-/-]) mice. Eight-wk-old *Apoe*[-/-] and *Apoe*[-/-]*Aff3ir-ORF2*[-/-] mice were fed a high-fat diet for 12 wk to induce atherosclerosis (*Figure 3A*). *En face* Oil-Red O staining indicated that *Aff3ir-ORF2* deficiency accelerated the development of atherosclerosis in the entire aorta, AA, and TA. (*Figure 3B*

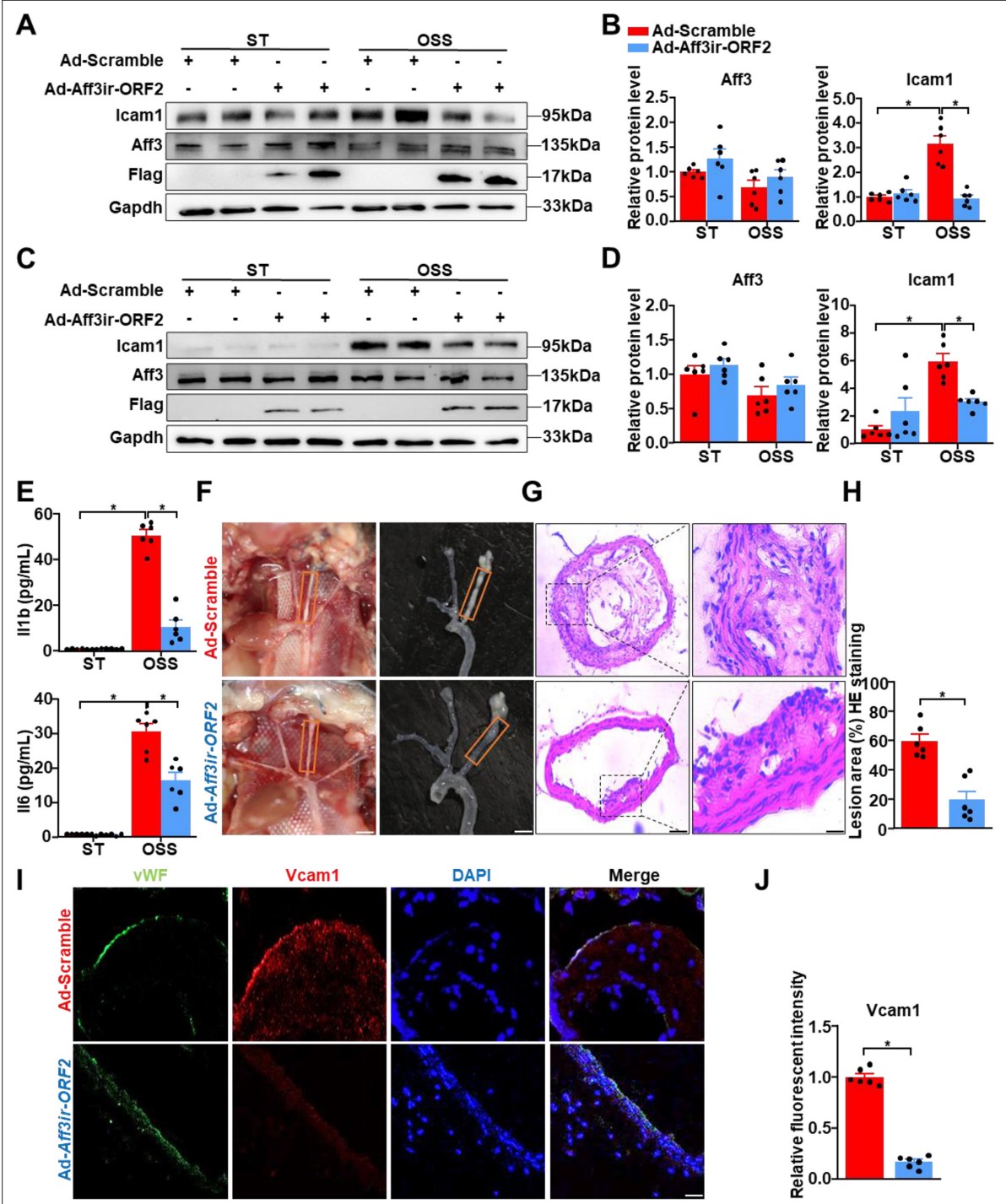

**Figure 2.** *Aff3ir-ORF2* overexpression alleviates disturbed flow-induced inflammation and atherosclerosis. (**A–B**) Mouse embryonic fibroblasts (MEFs) isolated from C57BL/6 mice were infected with indicated adenoviruses (Ad-Scramble or Ad-*Aff3ir-ORF2*) for 48 h and then exposed to static (ST) or oscillatory shear stress (OSS, 0.5±4 dyn/cm², 1 Hz) for another 6 hr. Western blot analysis of the indicated proteins and quantification of their relative expression levels are shown. The protein levels were normalized to *Gapdh* and the relative expression values were compared to MEFs infected with Ad-Scramble and treated with ST. Data are presented as mean ± SEM (n=6 independent experiments). *p<0.05, two-way ANOVA with Tukey post-test. (**C–D**) Human umbilical vein endothelial cells (HUVECs) were infected with Ad-Scramble or Ad-*Aff3ir-ORF2* for 48 h and then exposed to ST or OSS for an additional 6 hr. Western blot analysis of the indicated proteins and quantification of their relative expression levels are shown. The protein levels were normalized to *Gapdh* and the relative expression values were compared to HUVECs infected with Ad-Scramble and treated with ST. Data are presented as mean ± SEM (n=6 independent experiments). *p<0.05, two-way ANOVA with Tukey post-test. (**E**) MEFs were infected with Ad-Scramble or Ad-*Aff3ir-ORF2* for 48 h and then exposed to ST or OSS for another 6 hr. The concentration of *Il6* and *Il1b* in cell culture medium were detected

*Figure 2 continued on next page*

*Figure 2 continued*

with ELISA. The relative cytokine levels are relative to MEFs infected with Ad-Scramble and treated with ST. Data are presented as mean ± SEM (n=6 independent experiments). *p<0.05, two-way ANOVA with Tukey post-test. (F–J) Eight-wk-old male *Apoe*−/− mice were subjected to partial ligation of the carotid artery along with 10 μL of adenovirus suspension at 1×10⁸ transducing units (TU)/mL was instilled into the left carotid artery (LCA). The mice were then fed high-fat diet for 4 wk. (F) Arterial tissues were isolated to examine the atherosclerotic lesions. Scale bar, 2 mm. (G, H) LCAs were sectioned for hematoxylin and eosin staining. Quantification of the lesion area in LCAs was shown. Scale bar, 25 μm. Data are presented as mean ± SEM (n=6 mice per group). *p<0.05, unpaired two-tailed *t*-test. (I, J) Immunofluorescence staining for vWF, *Vcam1*, and DAPI in the LCAs, and quantification of the relative fluorescent intensity of *Vcam1*. The immunofluorescence intensity of *Vcam1* was normalized to DAPI, and the relative expression values were compared to that of the Ad-Scramble group. Scale bar, 50 μm. Data are presented as mean ± SEM (n=6 mice per group). *p<0.05, unpaired two-tailed *t*-test.

The online version of this article includes the following source data and figure supplement(s) for figure 2:

**Source data 1.** The table summarizes the data in the statistical graph for *Figure 2B, D, E, H and J*.

**Source data 2.** The original file of the full raw uncropped, unedited polyacrylamide gels for *Figure 2A*.

**Source data 3.** Figures with the uncropped polyacrylamide gels with the relevant bands clearly labeled for *Figure 2A*.

**Source data 4.** The original file of the full raw uncropped, unedited polyacrylamide gels for *Figure 2C*.

**Source data 5.** Figures with the uncropped polyacrylamide gels with the relevant bands clearly labeled for *Figure 2C*.

**Figure supplement 1.** *Aff3ir-ORF2* overexpression alleviates disturbed flow-induced inflammation in endothelial cells (ECs).

**Figure supplement 1—source data 1.** The table summarizes the data in the statistical graph for *Figure 2—figure supplement 1A, B and D*.

*and C*). Furthermore, *Aff3ir-ORF2* deletion increased the lesion area and lipid deposition in the aortic roots of *Apoe*-/- mice without altering the collagen fiber content (*Figure 3D and E*). Similar results were observed in distributing arteries (LCAs) (*Figure 3F and G*). Given that the expression of adhesion proteins, such as *Vcam1* in ECs is crucial for monocyte infiltration into plaques (*Kobiyama and Ley, 2018*), we assessed *Vcam1* expression in the aortic roots of these mice. We found that *Aff3ir-ORF2* deletion increased *Vcam1* expression in the aortic roots of *Apoe*-/- mice (*Figure 3H, I*), indicating that the atherogenic effects of *Aff3ir-ORF2* deletion may result from endothelial inflammation. Additionally, there were no significant differences between the two groups in body weight or triglyceride, total cholesterol, LDL cholesterol, and HDL cholesterol levels (*Figure 3—figure supplement 2A, B*), indicating that the atherogenic effect of *Aff3ir-ORF2* silencing is unlikely to be related to lipid metabolism. Taken together, these results indicate that *Aff3ir-ORF2* deficiency aggravates inflammation and atherosclerotic lesions in mice.

## *Aff3ir-ORF2* mitigates disturbed shear stress-induced inflammation by interacting with *Irf5* and retaining it within the cytosol

To explore the mechanism by which *Aff3ir-ORF2* mitigates atherogenesis, we performed RNA sequencing (RNA-seq) on MEFs from WT and *Aff3ir-ORF2*-/- mice. Expression of all differentially expressed genes in the *Supplementary file 3*. The Principal component analysis plot depicted a clear clustering of WT versus *Aff3ir-ORF2*-/- samples (*Figure 4A*). We identified 1167 upregulated and 310 downregulated genes in the *Aff3ir-ORF2*-/- group, with a criterion of 1.5-fold change and p<0.05 (*Figure 4B*, *Figure 4—figure supplement 1A*). All the differentially expressed genes were subjected to bioinformatics enrichment analysis using Gene Ontology (GO) databases. GO analysis showed that these genes were mainly enriched in processes, including leukocyte cell-cell adhesion, regulation of cell−cell adhesion, and leukocyte activation involved in immune response (*Figure 4C*), which is highly consistent with the phenotypes observed in *Aff3ir-ORF2*-/- mice. To further investigate the functional features of these differentially expressed genes in the context of the atherosclerotic microenvironment, we mapped the differential gene list onto the atherosclerosis-related gene dataset (*Rouillard et al., 2016*), resulting in 363 overlapping genes. GO analysis of these genes revealed enrichment in processes related to cell−cell adhesion and leukocyte activation involved in immune response (*Figure 4—figure supplement 1B*), which is highly consistent with the observed effects of *Aff3ir-ORF2* on *Vcam1* expression.

To further identify the upstream transcriptional regulators of these genes, we used the list of differentially expressed genes from the RNA-seq data to predict upstream transcription factors using the ChEA3 database (*Keenan et al., 2019*). Then, the top 20 transcription factors obtained from the ChEA3 database were mapped to the atherosclerotic disease-related gene list in the Disgenet

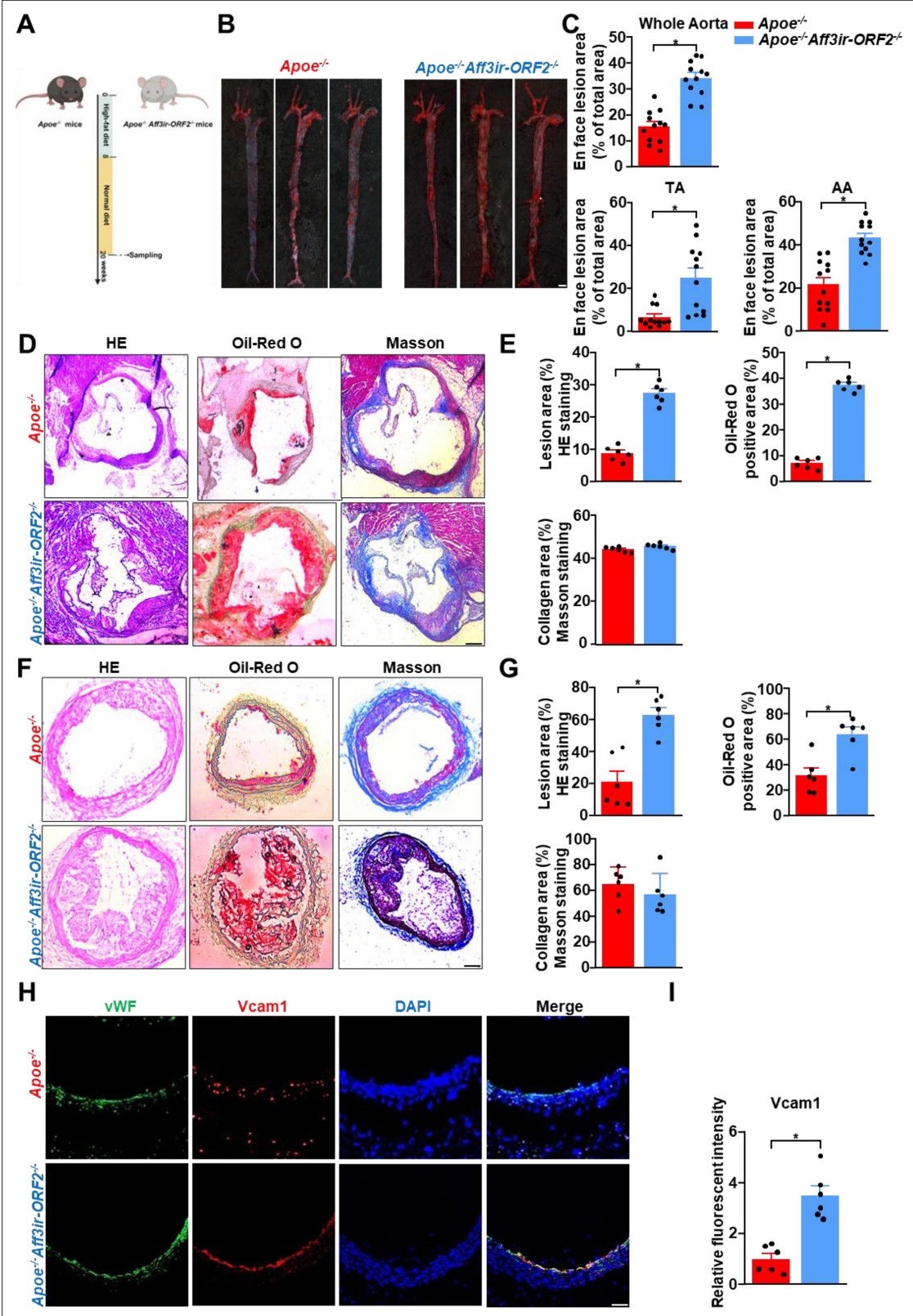

**Figure 3.** *Aff3ir-ORF2* deletion aggravates inflammation and atherosclerotic lesions in *Apoe⁻/⁻* mice. Eight-wk-old male *Apoe⁻/⁻* and *Apoe⁻/⁻Aff3ir-ORF2⁻/⁻* mice were fed a high-fat diet for 12 wk. Arterial tissues and aortic roots were isolated to examine atherosclerotic lesions. (**A**) Schematic of experimental strategy. (**B**) Representative images of en face Oil-Red O staining of the aortas. Scale bar, 4 mm. (**C**) Quantification of the plaque area in the whole aorta, aortic arch (AA), and thoracic aorta (TA). Data are presented as mean ± SEM (n=12 mice per group). *p<0.05, unpaired two-tailed *t*-test. (**D**) Oil-Red

*Figure 3 continued on next page*

*Figure 3 continued*

O, hematoxylin and eosin (HE), and Masson staining of the aortic roots. Scale bars, 500 μm. (**E**) Quantification of plaque size, Oil-Red O-positive area, and collagen fiber content in aortic root sections. Data are presented as mean ± SEM (n=6 mice per group). *p0.05, unpaired two-tailed *t*-test. (**F**), left carotid arteries (LCAs) were sectioned and stained with Oil-Red O, HE, and Masson's trichrome. Scale bars, 500 μm. (**G**) Quantification of plaque size, Oil-Red O-positive area, and collagen fiber content in the LCA sections. Data are presented as mean ± SEM (n=6 mice per group). *p<0.05, unpaired two-tailed *t*-test. (**H**) Representative immunofluorescence images of vWF, *Vcam1*, and DAPI in the aortic roots. Scale bar, 500 μm. (**I**), Quantification of the relative fluorescence intensity of *Vcam1*. The immunofluorescence intensity of *Vcam1* was normalized to that of DAPI, and the relative expression values were compared to that of the *Apoe*[-/-] group. Data are presented as mean ± SEM (n=6 mice per group). *p<0.05, unpaired two-tailed *t*-test.

The online version of this article includes the following source data and figure supplement(s) for figure 3:

**Source data 1.** The table summarizes the data in the statistical graph for *Figure 3A, E, G and I*.

**Figure supplement 1.** *Aff3ir-ORF2* is dispensable for physiological blood pressure maintenance.

**Figure supplement 1—source data 1.** The table summarizes the data in the statistical graph for *Figure 3—figure supplement 1C, D, E and F*.

**Figure supplement 1—source data 2.** The original file of the full raw uncropped, unedited polyacrylamide gels for *Figure 3—figure supplement 1B*.

**Figure supplement 1—source data 3.** Figures with the uncropped polyacrylamide gels with the relevant bands clearly labeled for *Figure 3—figure supplement 1B*.

**Figure supplement 1—source data 4.** The original file of the full raw uncropped, unedited agarose gels for *Figure 3—figure supplement 1A*.

**Figure supplement 1—source data 5.** Figures with the uncropped agarose gels with the relevant bands clearly labeled for *Figure 3—figure supplement 1A*.

**Figure supplement 2.** *Aff3ir-ORF2* is dispensable for lipid metabolism maintenance.

**Figure supplement 2—source data 1.** The table summarizes the data in the statistical graph for *Figure 3—figure supplement 2A, B*.

database (*Piñero et al., 2017*). Interferon regulatory factor 5 (*Irf5*) and *Irf8* were identified as key upstream regulators (*Figure 4D*). *Irf5* and *Irf8*, which are members of the same family of transcription factors originally implicated in interferon production, have been identified as critical regulators of the inflammatory response and contribute to the pathogenesis of various inflammatory diseases (*Almuttaqi and Udalova, 2019*; *Salem et al., 2020*). However, their potential roles in disturbed shear stress-induced inflammation remain unclear. We speculated that *Aff3ir-ORF2* interacts with *Irf5* and/or *Irf8*. Coimmunoprecipitation assays indicated that endogenous *Aff3ir-ORF2* could bind to both *Irf5* and *Irf8* (*Figure 4E*). To determine which transcription factor mediates the inflammatory effects of *Aff3ir-ORF2* deficiency, we silenced *Irf5* and *Irf8* in WT and *Aff3ir-ORF2*[-/-] MEFs exposed to disturbed flow. Notably, silencing *Irf5*, but not *Irf8*, blunted the upregulation of inflammatory genes, including *Icam1*, *Vcam1*, *Il6*, and *Il1b* (*Figure 4F*), suggesting that *Irf5* was the predominant factor mediating the anti-inflammatory effects of *Aff3ir-ORF2* in the context of disturbed shear stress. Consistently, we found that *Irf5* silencing significantly inhibited the upregulation of *Icam1* protein levels induced by *Aff3ir-ORF2* deficiency under disturbed shear stress (*Figure 4G and H*). In addition, neither *Irf5* nor *Irf8* expression levels were affected by *Aff3ir-ORF2* deficiency (*Figure 4—figure supplement 1C*). However, we found that *Aff3ir-ORF2* deficiency significantly increased the expression of *Irf5*-targeted genes (predicted by the ChEA3 database), including *Icam1*, *Ccl5*, and *Cxcl10* (*Figure 4—figure supplement 1D*). Notably, the protein level of *Irf5* was not significantly affected by disturbed shear stress (*Figure 4G and H*). Consistently, the mRNA levels of *Irf5* were previously reported to be barely changed in the context of disturbed shear stress (*Deng et al., 2025*) (GSE276195, *Figure 4—figure supplement 1E*) or the atherosclerotic environment (*Li et al., 2025*) (GSE222583, *Figure 4—figure supplement 1F*). Given that the transcriptional activity of *Irf5* depends on its nuclear translocation (*Lv et al., 2024*), we next explored whether *Aff3ir-ORF2* affects the subcellular localization of *Irf5*. Subcellular fractionation assays indicated that *Irf5* was predominantly localized in the cytoplasm under static conditions, but exhibited obvious nuclear localization when exposed to disturbed shear stress (*Figure 4I and J*). While the total expression of *Irf5* was barely affected by *Aff3ir-ORF2* deficiency or overexpression, nuclear localization of *Irf5* increased with *Aff3ir-ORF2* deficiency (*Figure 4I and J*). To further ascertain the role of *Aff3ir-ORF2* in regulating the transcriptional activity of *Irf5*, we performed a luciferase reporter assay (*Qiao et al., 2022*). *Aff3ir-ORF2* overexpression significantly decreased the transcriptional activity of *Irf5* (*Figure 4K*). In summary, these results suggested that *Aff3ir-ORF2* acts as an endogenous inhibitor of *Irf5* and exerts anti-inflammatory effects by retaining *Irf5* in the cytosol.

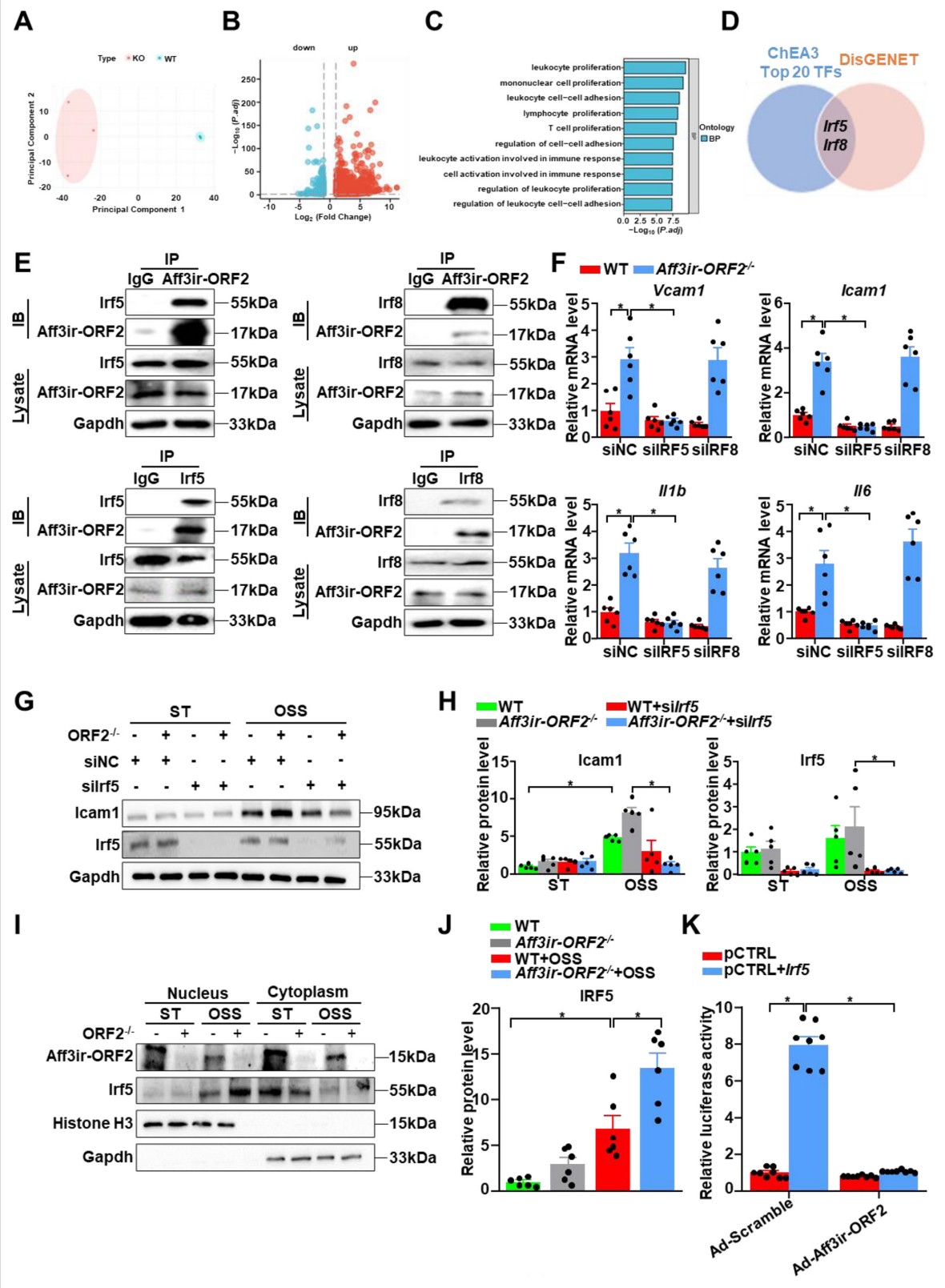

**Figure 4.** *Aff3ir-ORF2* mitigates disturbed shear stress-induced inflammation by interacting with *Irf5* and retaining it within the cytosol. (**A–D**) Mouse embryonic fibroblasts (MEFs) were isolated from wild-type (WT) and *Aff3ir-ORF2-/-* mice. (**A**) Principal component analysis (PCA) analysis of RNA-seq data to visualize sample-to-sample variation. (**B**) Volcano map showing mRNA profiles of WT and *Aff3ir-ORF2-/-* MEFs (n=3). (**C**) Gene Ontology enrichment pathway analysis of the differentially expressed genes. (**D**) Venn diagrams of the top 20 transcription factors from the ChEA3 and DisGENET analysis

*Figure 4 continued on next page*

*Figure 4 continued*

related to atherosclerosis. (**E**) Immunoprecipitation performed using antibodies against *Aff3ir-ORF2*, *Irf5*, and *Irf8*. n=3 independent experiments. (**F–H**), WT and *Aff3ir-ORF2$^{-/-}$* MEFs were subjected to silence of Control (siNC), *Irf5* (si*Irf5*), or *Irf8* (si*Irf8*) with siRNAs for 24 hr, followed by exposure to static (ST) or oscillatory shear stress (OSS, 0.5±4 dyn/cm$^2$, 1 Hz) for another 6 hr. (**F**) RT-PCR analysis of the mRNA levels of *Vcam1*, *Icam1*, *Il6*, and *Il1b*. The relative expression values were compared to WT MEFs transfected with siNC and treated with ST. Data are mean ± SEM (n=6 independent experiments). *p<0.05, two-way ANOVA with Tukey post-test. (**G–H**), Representative western blots of *Irf5* and *Icam1* expression. Data are mean ± SEM (n=5 independent experiments). *p<0.05, two-way ANOVA with Tukey post-test. (**I–J**), WT, and *Aff3ir-ORF2$^{-/-}$* MEFs were exposed to ST or OSS for 6 hr. Nuclear and cytoplasmic proteins were extracted from the cells. Representative western blots of the indicated proteins and quantification of *Irf5* expression in nucleus are shown. The expression of these proteins was relative to the level of nuclear *Irf5* in ST-treated WT MEFs. Data are mean ± SEM (n=6 independent experiments). *p<0.05, one-way ANOVA with Tukey post-test. (**K**) HEK293 cells were transfected with the firefly luciferase reporter plasmid containing the *Irf5*-responsive ZNF217 promoter along with a β-galactosidase reporter plasmid for 24 hr. Cells were infected with the indicated adenoviruses (Ad-Scramble or Ad-*Aff3ir-ORF2*) for 24 h. Promoter activity was measured using luciferase, which was normalized to β-gal. Data are mean ± SEM (n=6 independent experiments). *p<0.05, two-way ANOVA with Tukey post-test.

The online version of this article includes the following source data and figure supplement(s) for figure 4:

**Source data 1.** The table summarizes the data in the statistical graph for *Figure 4F, H, J and K*.

**Source data 2.** The original file of the full raw uncropped, unedited polyacrylamide gels for *Figure 4E*.

**Source data 3.** Figures with the uncropped polyacrylamide gels with the relevant bands clearly labeled for *Figure 4E*.

**Source data 4.** The original file of the full raw uncropped, unedited polyacrylamide gels for *Figure 4G*.

**Source data 5.** Figures with the uncropped polyacrylamide gels with the relevant bands clearly labeled for *Figure 4G*.

**Source data 6.** The original file of the full raw uncropped, unedited polyacrylamide gels for *Figure 4I*.

**Source data 7.** Figures with the uncropped polyacrylamide gels with the relevant bands clearly labeled for *Figure 4I*.

**Figure supplement 1.** *Aff3ir-ORF2* deficiency significantly increased the expression of *Irf5*-targeted genes.

**Figure supplement 1—source data 1.** Figures with the uncropped agarose gels with the relevant bands clearly labeled for *Figure 4—figure supplement 1C, D, E and F*.

## *Irf5* knockdown prevents the aggravation of atherosclerosis induced by ORF2 deficiency

Next, we investigated the role of *Irf5* in disturbed flow-induced atherosclerosis in vivo and whether it mediates the atherogenic phenotype associated with *Aff3ir-ORF2* deficiency. *Apoe$^{-/-}$* and *Apoe$^{-/-}$Aff3ir-ORF2$^{-/-}$* mice were subjected to partial ligation surgery in the LCAs and intravascularly infected with lentiviruses expressing either *Irf5*-specific shRNA (lenti-sh*Irf5*) or Scramble shRNA (lenti-shScramble). En-face immunofluorescence staining confirmed successful *Irf5* deletion in the left carotid artery (*Figure 5—figure supplement 1A*). After a 4 wk high-fat diet challenge, *Irf5* deletion resulted in an approximately 60% reduction in plaque area in the LCAs of *Apoe$^{-/-}$* mice (*Figure 5A and B*). In addition, *Irf5* deletion attenuated endothelial activation, as evidenced by reduced *Vcam1* expression in the endothelium of LCAs (*Figure 5C and D*). Notably, although *Apoe$^{-/-}$Aff3ir-ORF2$^{-/-}$* mice exhibited an increased plaque area in the LCAs compared to *Apoe$^{-/-}$* mice, *Irf5* deletion almost completely abolished these differences, reducing both the plaque area and *Vcam1* expression in the endothelium of the LCAs (*Figure 5A–D*). These findings provide in vivo evidence that *Aff3ir-ORF2* deficiency-induced atherosclerosis is mediated by endothelial *Irf5*.

## Endothelial-specific *Aff3ir-ORF2* supplementation ameliorates EC activation and atherosclerosis in mice

Given the significant anti-inflammatory effects of *Aff3ir-ORF2* on endothelial activation and atherosclerosis, we explored the potential use of gene therapy targeting *Aff3ir-ORF2* to treat atherosclerosis. Endothelial-specific *Aff3ir-ORF2* overexpression was achieved using an EC-enhanced AAV-mediated CRISPR/Cas9 genome-editing system controlled by an EC-specific ICAM2 promoter as we previously reported (*Li et al., 2024*; *Swiech et al., 2015*). *Apoe$^{-/-}$* mice infected with AAV-ICAM2-Control or AAV-ICAM2-*Aff3ir-ORF2* were fed a high-fat diet for 12 wk (*Figure 6A*). En-face immunofluorescence staining confirmed successful *Aff3ir-ORF2* overexpression in ECs (*Figure 6—figure supplement 1A–B*). Endothelial-specific *Aff3ir-ORF2* overexpression had a minimal effect on triglycerides, total cholesterol, LDL cholesterol, and HDL cholesterol levels in the plasma of mice (*Figure 6—figure supplement 1C*). However, compared to the negative control, endothelial-specific *Aff3ir-ORF2*

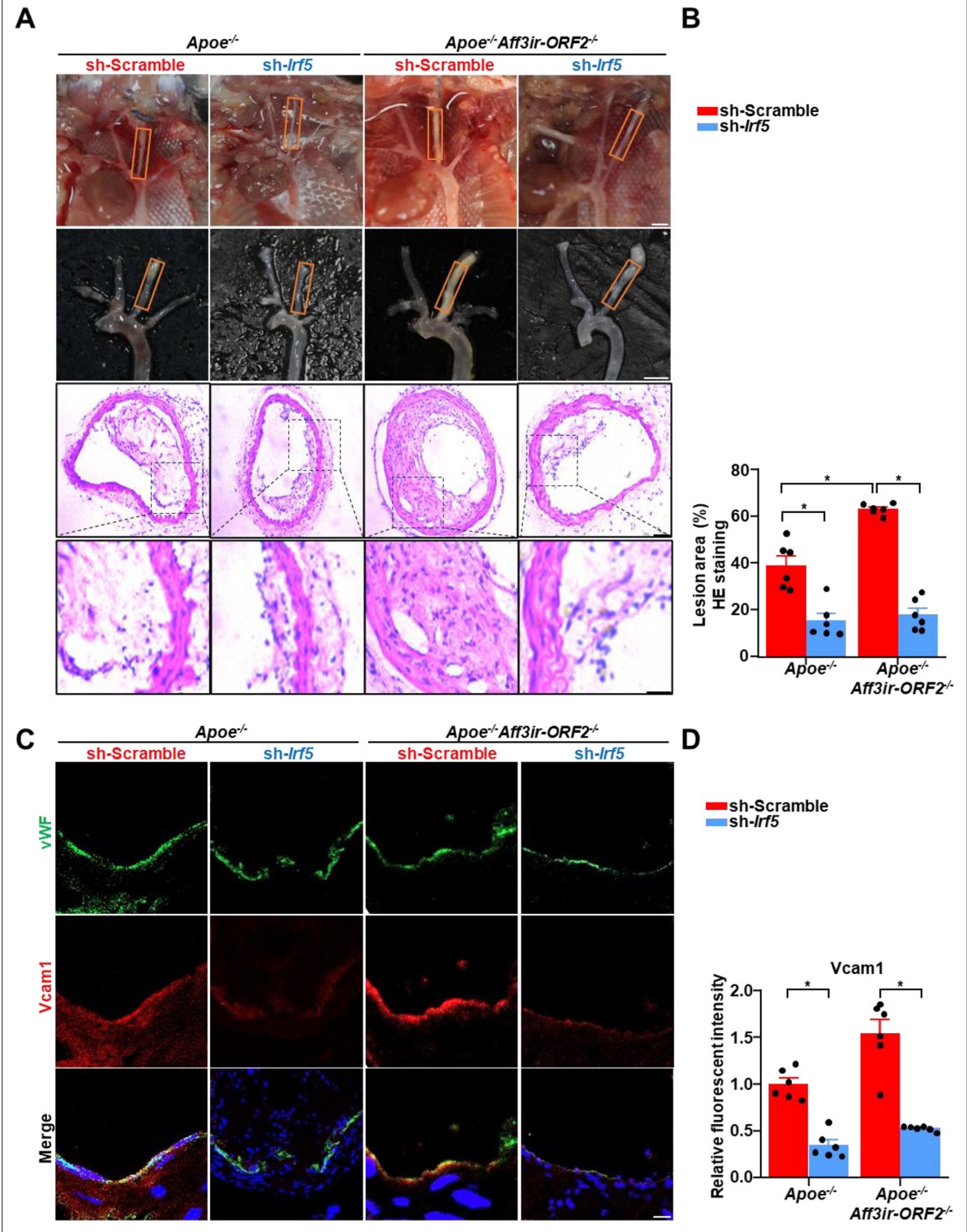

**Figure 5.** *Irf5* knockdown prevents the aggravation of atherosclerosis in *Aff3ir-ORF2* deficient mice. Eight-wk-old male *Apoe^-/-* mice were subjected to partial ligation of the left carotid artery (LCA) along with 10 μL of lentivirus suspension at $1×10^8$ transducing units (TU)/mL was instilled into the LCA. The mice were then fed a high-fat diet for 4 wk. (**A–B**) Arterial tissues were isolated to examine the atherosclerotic lesions. LCAs were sectioned for hematoxylin and eosin staining. Quantification of the lesion area in LCAs was shown. Data are mean ± SEM (n=6 mice per group). *$p<0.05$, two-way ANOVA with Tukey post-test. Scale bar: 2 mm for gross images, 25 μm for staining images. (**C–D**) Immunofluorescence staining for vWF, *Vcam1*, and DAPI in the LCAs and quantification of the relative fluorescent intensity of *Vcam1*. Scale bar, 50 μm. The immunofluorescence intensity of *Vcam1*

*Figure 5 continued on next page*

*Figure 5 continued*

was normalized to DAPI, and the relative expression values were compared to that of the group of *Apoe*-/- mice infected with Ad-scramble. Data are presented as mean ± SEM (n=6 mice per group). *p<0.05, two-way ANOVA with Tukey post-test.

The online version of this article includes the following source data and figure supplement(s) for figure 5:

**Source data 1.** The table summarizes the data in the statistical graph for *Figure 5B and D*.

**Figure supplement 1.** *Irf5* deletion in the left carotid artery successfully.

**Figure supplement 1—source data 1.** Figures with the uncropped agarose gels with the relevant bands clearly labeled for *Figure 5—figure supplement 1B*.

overexpression significantly reduced the Oil-red O-positive lesion area in the whole aortas of *Apoe*-/- mice (19 ± 5% vs 54 ± 8%) (*Figure 6B and C*). Moreover, endothelial-specific *Aff3ir-ORF2* over-expression reduced the lesion area and lipid deposition in the aortic roots of *Apoe*-/- mice without altering the collagen fiber content (*Figure 6D and E*). In addition, *Aff3ir-ORF2* overexpression effectively suppressed *Vcam1* expression in the endothelium of the aortic roots of *Apoe*-/- mice (*Figure 6F and G*). Collectively, these results suggest that supplementation with *Aff3ir-ORF2* was effective in preventing atherosclerosis development.

## Discussion

Endothelial activation is a critical initial event in the development of atherosclerosis, and emerging evidence suggests that targeting disturbed shear stress-induced endothelial activation is a promising therapeutic strategy. In the present study, we elucidated the role of the novel nested gene-encoded protein, *Aff3ir-ORF2*, in sensing disturbed shear stress. Moreover, we demonstrated that *Aff3ir-ORF2* acts as an endogenous inhibitor of *Irf5*, a key regulator of the inflammatory response, thereby exerting potent anti-inflammatory and anti-atherogenic effects (*Figure 7*).

Using three mouse models (global *Aff3ir-ORF2* knockout, locally *Aff3ir-ORF2* endothelial expression, and endothelial-specific *Aff3ir-ORF2* overexpression), we demonstrated that *Aff3ir-ORF2* exerted potent anti-inflammatory and anti-atherosclerosis effects in *Apoe*-/- mice. Notably, while *Aff3ir-ORF2* knockout increased *Vcam1* expression in the endothelium and enlarged the plaque area in the aortic roots, it had a minimal effect on collagen deposition within the plaques. This discrepancy may be attributed to the differential expression patterns of *Irf5* across various cell types (*Roberts et al., 2024*). Phenotypically modulated vascular smooth muscle cells (VSMCs) within the fibrous cap produce extracellular matrix molecules critical for plaque composition and stabilization (*Bennett et al., 2016*). A previous study has shown minimal colocalization between *Irf5* and the VSMC marker, α-smooth muscle actin, in aortic root lesions of *Apoe*-/- mice (*Seneviratne et al., 2017*), indicating a relatively low *Irf5* expression level in VSMCs. Consequently, *Aff3ir-ORF2* likely exerts its anti-inflammatory effects primarily through endothelial *Irf5*. Consistently, endothelial-specific *Aff3ir-ORF2* overexpression reduced the aortic plaque area, but had minimal effects on collagen deposition. Our findings establish the potent anti-atherosclerotic role of *Aff3ir-ORF2* in early and advanced atherosclerosis mouse models. However, given the multiple critical roles of ECs throughout the initiation and progression of atherosclerosis, further investigations are needed to explore the potential role of *Aff3ir-ORF2* in other atherosclerotic processes, including endothelial-to-mesenchymal transition, plaque rupture, and atherothrombotic occlusion. Additionally, we found that disturbed shear stress transcriptional downregulated the expression of *Aff3ir*. However, the protein levels of *Aff3ir-ORF2*, but not those of *Aff3ir-ORF1*, were reduced by disturbed shear stress. Since both *Aff3ir-ORF1* and *Aff3ir-ORF2* are derived from *Aff3ir*, different translation mechanisms may be involved in the production of *Aff3ir-ORF1/2*.

In addition to ECs, various other cell types, particularly immune cells, play crucial roles in the progression of atherosclerotic plaques (*Wolf and Ley, 2019*). Although our results indicated the potent anti-inflammatory role of *Aff3ir-ORF2* in ECs, the potential contributions of other cell types may also be involved, which could be further elucidated using *Aff3ir-ORF2* tissue-specific knockout or over-expression mouse models. Macrophage polarization and inflammatory responses accelerate plaque development, leading to an increase in necrotic core and vulnerable plaques (*Krausgruber et al., 2011*). Elevated *Irf5* expression and nuclear localization have been observed in macrophages within

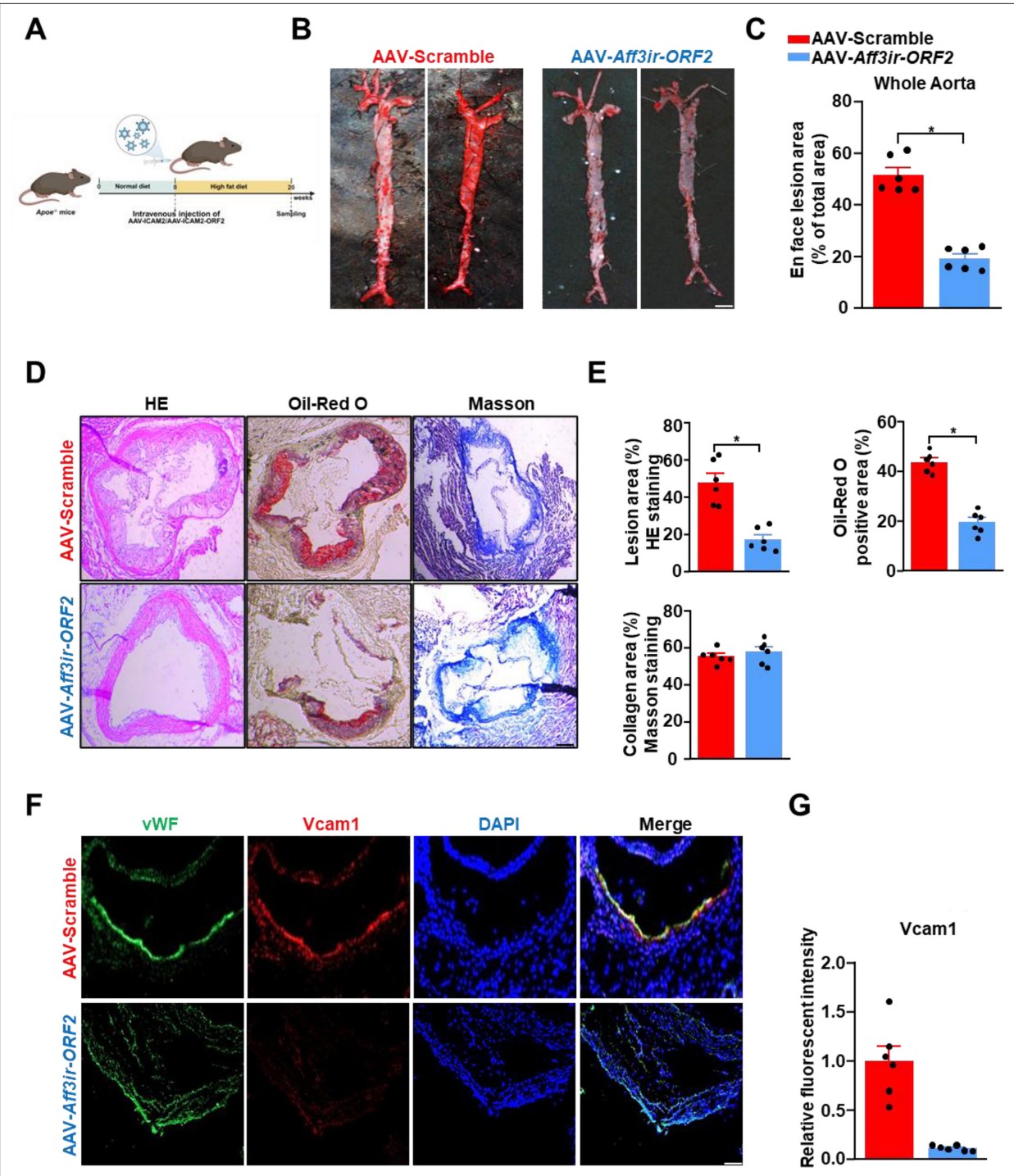

**Figure 6.** Endothelial-specific *Aff3ir-ORF2* supplementation alleviates EC activation and atherosclerosis in *Apoe⁻/⁻* mice. Eight-wk old *Apoe⁻/⁻* male mice were infused with the indicated adeno-associated virus (AAV) and then fed a high-fat diet for 12 wk. (**A**) Schematic of the experimental strategy. (**B**) Representative images of enface Oil-Red O staining of the aortas. Scale bar, 4 mm. (**C**) Quantification of the plaque area in the entire aortas. Data are presented as mean ± SEM (n=6 mice per group). *p<0.05, unpaired two-tailed *t*-test. (**D**) Hematoxylin and eosin (HE), Oil-Red O, and Masson staining of the aortic roots. Scale bars, 500 µm. (**E**) Quantification of plaque size, Oil-Red O-positive area, and collagen fiber content in aortic root sections. Data are presented as mean ± SEM (n=6 mice per group). *p<0.05, unpaired two-tailed *t*-test. (**F**) Representative immunofluorescence image of vWF, *Vcam1*, and DAPI in the aortic roots. Scale bar, 500 µm. (**G**) Quantification of the relative fluorescent intensity of *Vcam1*. The immunofluorescence intensity of *Vcam1* was normalized to that of DAPI, and the relative expression values were compared to that of the AAV-Scramble group. Data are presented as mean ± SEM (n=6 mice per group). *p<0.05, unpaired two-tailed *t*-test.

*Figure 6 continued on next page*

*Figure 6 continued*

The online version of this article includes the following source data and figure supplement(s) for figure 6:

**Source data 1.** The table summarizes the data in the statistical graph for *Figure 6C, E and G*.

**Figure supplement 1.** *Aff3ir-ORF2* overexpression in endothelial cells (ECs) successfully.

**Figure supplement 1—source data 1.** Figures with the uncropped agarose gels with the relevant bands clearly labeled for *Figure 6—figure supplement 1B and C*.

plaques of *Apoe*[-/-] mice (*Seneviratne et al., 2017*). *Irf5* has been demonstrated to drive macrophages towards a pro-inflammatory state, thereby affecting plaque stability (*Seneviratne et al., 2017*). Global or myeloid cell-specific deletion of *Irf5* stabilizes atherosclerotic plaques by suppressing the inflammatory phenotypes of macrophages (*Leipner et al., 2021*; *Seneviratne et al., 2017*). Despite these findings, the role of *Irf5* in shear stress-induced endothelial activation remains largely unknown. Our study provides evidence that disturbed shear stress is sufficient to induce *Irf5* nuclear translocation and activation in ECs. Furthermore, *Irf5* knockdown in ECs significantly reduced disturbed flow-induced

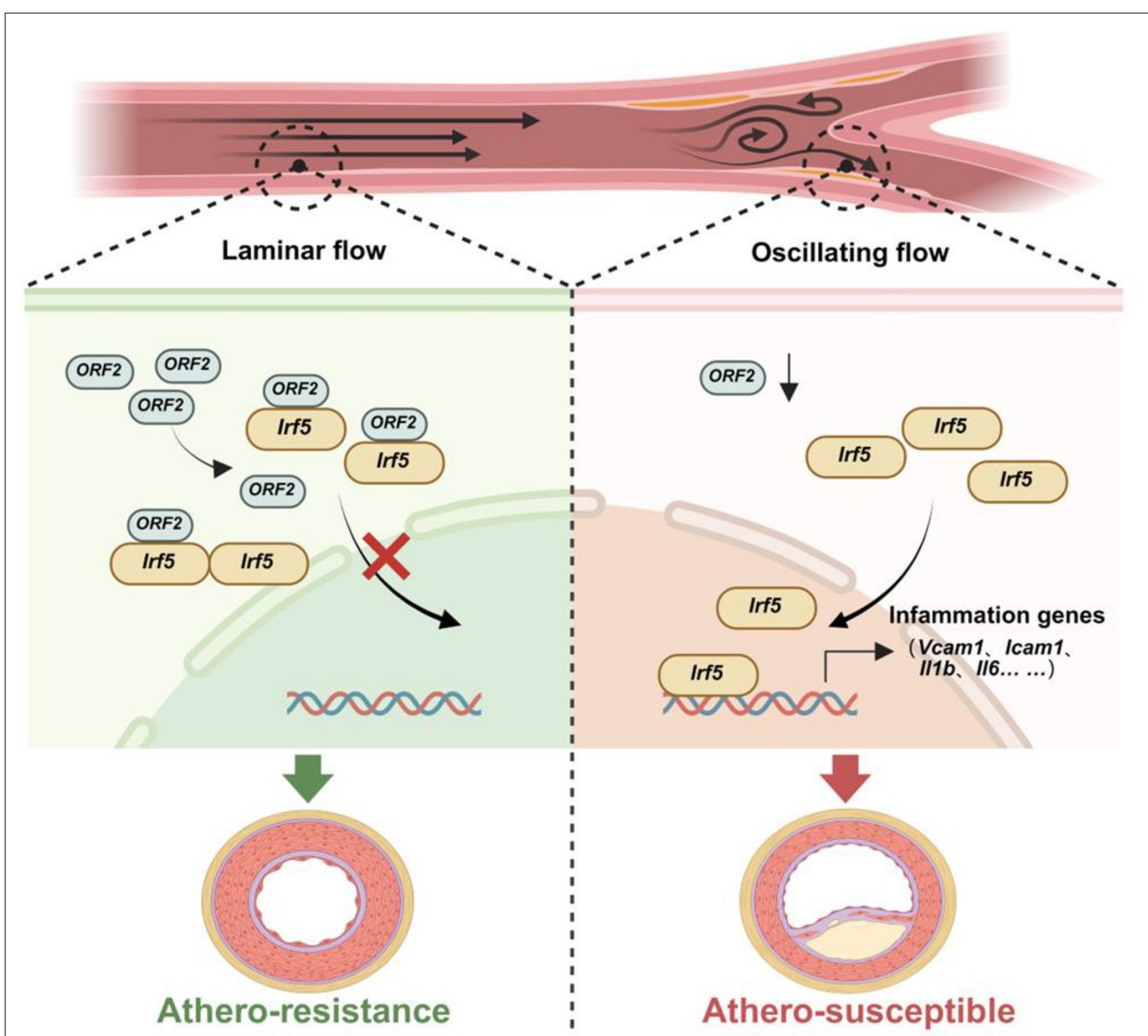

**Figure 7.** Schematic illustration of the *Aff3ir-ORF2/Irf5* cascade in disturbed flow-induced endothelial activation and atherosclerosis. Disturbed flow induced a down-regulation of *Aff3ir-ORF2*, which could interact with *Irf5* and promote the latter's retention in the cytoplasm, thereby boosting *Irf5*-dependent inflammatory pathways in endothelial cells and leading to atherogenesis.

plaque formation in LCAs. These findings suggest that targeting endothelial *Irf5* may be an effective strategy for combating the early stages of atherosclerosis.

Given the key role of *Irf5* in mediating inflammatory responses, it has been considered as an attractive therapeutic target, and various strategies have been developed to study and modulate its function (*Almuttaqi and Udalova, 2019*). For example, nanoparticle-delivered siRNA targeting *Irf5* in macrophages promotes inflammation resolution, improves infarct healing, and attenuates post-myocardial infarction remodeling (*Courties et al., 2014*). Additionally, manipulating *Irf5* protein levels through the E3 ubiquitin ligase, *Trim21*, has been explored as a strategy for modulating its activity (*Lazzari et al., 2014*). Given the crucial physiological role of *Irf5*, strategies aimed at suppressing its pathophysiological activation without altering basal levels may offer additional benefits. Our study introduces a novel approach to inhibit *Irf5* activation. We found that the novel nested gene-encoded protein, *Aff3ir-ORF2*, interacts with *Irf5*, leading to cytoplasmic retention and inactivation under disturbed shear stress conditions. Importantly, endothelium-specific supplementation with *Aff3ir-ORF2* effectively attenuated endothelial activation and reduced the atherosclerotic plaque area in *Apoe*-/- mice, suggesting that targeting endothelial *Irf5* activation with *Aff3ir-ORF2* holds promise for the treatment of atherosclerosis. Furthermore, as emerging studies have highlighted the substantial contributions of *Irf5* to autoimmune diseases (*Graham et al., 2006*), neuropathic pain (*Masuda et al., 2014*), obesity (*Dalmas et al., 2015*), and hepatic fibrosis (*Alzaid et al., 2016*), future research should investigate whether *Aff3ir-ORF2* has beneficial effects in these contexts.

## Conclusion
In conclusion, this study provides novel evidence that the disruption of *Aff3ir-ORF2* expression under disturbed flow promotes endothelial inflammatory responses and atherosclerosis. *Aff3ir-ORF2* serves as an endogenous inhibitor of *Irf5* by binding to *Irf5* and preventing its nuclear translocation. Supplementation with endothelial *Aff3ir-ORF2* may be a promising therapeutic strategy for treating atherosclerosis.

# Materials and methods
## Animals
C57BL/6 and Apolipoprotein E-null (*Apoe*-/-) mice were purchased from the Experimental Animal Centre of the Military Medical Science Academy (Beijing, China). *Aff3ir-ORF2*-heterozygote (*Aff3ir-ORF2*+/-) mice were acquired from Dr. Lingfang Zeng's Laboratory at King's College London. Briefly, gRNAs targeting the mouse *Aff3ir-ORF2* locus (gRNA1: GCAACCCACGGAGTTGCAGTTGG; gRNA2: GTCA TTAACTCCTTTAATATAGG; gRNA3: TGCAACTCCGTGGGTTGCTGTGG; gRNA4: GACCACACATAA CAGTGAATAGG) and Cas9 mRNA were co-injected into fertilized mouse eggs to generate targeted knockout offspring. F0 founder animals were identified by PCR followed by sequence analysis, and then bred to WT mice to test germline transmission and produce F1 animals. Heterozygous targeted mice were intercrossed to generate homozygous targeted mice. The genotyping primers used were: 5'-GGAAAGACCACAGAATCAATGACA-3', 5'-AACATTGCTATACCCCACTATA-3'. To generate *Apoe* and *Aff3ir-ORF2* double knockout mice (*Apoe*-/-*Aff3ir-ORF2*-/-), *Apoe*-/- mice were crossed with *Aff3ir-ORF2*-/- mice. The animals were maintained at 21 ± 1°C under a 12 hr light/dark cycle (lights on at 07:00, lights off at 19:00) with ad libitum access to water and standard chow unless specified otherwise. This study adhered to the *Guide for the Care and Use of Laboratory Animals* of the US National Institutes of Health (NIH Publication No. 85–23, revised 2011). All study protocols were approved by the Institutional Animal Care and Use Committee of Tianjin Medical University.

## Carotid artery partial ligation surgery
The surgery was performed as we previously described (*Yang et al., 2021*). Briefly, mice were anesthetized with isoflurane (2–3%). A ventral midline incision (4–5 mm) was made in the neck and the left carotid artery was exposed through blunt dissection of subcutaneous fat and muscle tissue. The left external carotid, internal carotid, and occipital arteries were ligated with a 6–0 silk suture, leaving the superior thyroid artery intact. For adenovirus and lentivirus infection studies, adenovirus (Ad-ORF2 or Ad-Scramble) or lentivirus (lenti-shRNA-*Irf5* or lenti-shRNA-Scramble) was introduced into the lumen of the left carotid artery and kept inside for 40 min. After infection, the adenovirus or lentivirus was

released, and blood flow to the common carotid artery was restored. Mice were fed with a high-fat diet (TD88137, ENVIGO, USA) immediately after surgery and continued for 4 wk.

## Endothelial *Aff3ir-ORF2* overexpression in mice

Endothelial-specific adeno-associated virus (AAV)-mediated CRISPR/Cas9 shuttle plasmid was constructed by Cell & Gene Therapy (Shanghai, China) as previously reported (*Wang et al., 2016*). *Apoe*$^{-/-}$ mice received a single tail vein injection of recombinant AAV containing an endothelial-specific human ICAM-2 promoter driving *Aff3ir-ORF2* overexpression (AAV-*Aff3ir-ORF2*) or a control empty vector (AAV-Scramble), with a dose of $1\times10^{11}$ viral genomes in a 200 µL vol of sterile PBS. Subsequently, the mice were fed with a high-fat diet (TD88137, ENVIGO, USA) for 3 mo.

## Oil-Red O staining for atherosclerotic plaques in mouse aorta

The *Apoe*$^{-/-}$, *Apoe*$^{-/-}$*Aff3ir-ORF2*$^{-/-}$, and EC-specific *Aff3ir-ORF2* overexpression mice were anesthetized by inhalation of 2% isoflurane and euthanized by cervical dislocation. The aortas were dissected in 1x PBS and opened to expose the atherosclerotic plaques. After fixation in 4% formaldehyde for 1 hr at 4 °C, the tissues were rinsed in water for 10 min, followed by 60% isopropanol. The aortas were then stained with Oil-Red O for 30 min with gentle shaking, rinsed again in 60% isopropanol, and subsequently rinsed in water three times. The samples were mounted on wax with the endothelial surface facing upwards. Images were captured using an HP Scanjet G4050. Plaque areas were quantified using NIH ImageJ software by calculating the plaque area relative to the total vascular area.

## Immunofluorescence staining

MEFs slides or frozen sections were fixed in 4% paraformaldehyde for 30 min, then permeabilized in 0.1% Triton X-100 (in PBS) and blocked with 1% bovine serum albumin for 30 min at room temperature. Sections were incubated overnight at 4 °C with primary antibodies (1:100). *Aff3ir-ORF2* (Cat. No. C0302HL300-4) antibody was from GenScript (Piscataway, NJ, USA). The vWF (Cat. No. ab11713) and VE-Cadherin (Cat. No. ab33168) antibodies were obtained from Abcam (Cambridge, UK). *Vcam1* (Cat. No. sc-13160) antibody was from Santa Cruz Biotechnology (Santa Cruz, CA, USA). Following primary antibody incubation, sections were treated with Alexa Fluor 488- or Alexa Fluor 594-conjugated secondary antibodies (1:200, Thermo Fisher Scientific, Grand Island, NY, USA) at room temperature for 1 hr. Slides were then mounted with DAPI-containing mounting medium. Antibody specificity and target staining authenticity were verified using negative controls. Immunofluorescence micrographs were acquired using a Leica confocal laser scanning microscope. Representative images were randomly selected from each group.

## Histological analysis of atherosclerotic lesions

Harvested carotid arteries and cross-sections of aortic roots were fixed in 4% paraformaldehyde and embedded in optimal cutting temperature compound (OCT). OCT-embedded tissues were sectioned at a thickness of 7 µm. Slides were immersed in 1x PBS for 5 min to remove OCT, and subsequently stained with Oil-Red O, hematoxylin and eosin (HE), and Masson's trichrome stain to assess lipid accumulation, lesion area, and collagen deposition, respectively (*Li et al., 2019*). Images were acquired using microscopy.

## Quantification of plasma lipid levels

Blood samples were obtained via cardiac puncture, rinsed with heparin, and collected in 1.5 mL Eppendorf tubes. Total plasma cholesterol, triglycerides, LDL cholesterol, and HDL cholesterol levels were measured enzymatically using an automated clinical chemistry analyzer kit (Biosino Biotech, Beijing, China).

## Cell culture, transfection, and shear stress experiments

Mouse Embryonic Fibroblasts (MEFs) were obtained and cultured as previously described (*Ferreira and Hein, 2023*). Cell passages 4–7 were used in all experiments. MEFs were cultured in the Dulbecco's Modified Eagle's Medium (DMEM) supplemented with 10% FBS, penicillin (100 U/mL), and streptomycin (100 µg/mL). Cells were incubated at 37 °C in a humidified environment containing 5% $CO_2$ and grown to 70–80% confluence before treatment.

Small interfering RNA against *Irf5* or *Irf8* were synthesized from General Biosystems (Hefei, China). The sequences of siRNAs are shown in the **Supplementary file 1**. The MEFs were passaged to six-well plates and transfected with 20 nmol/L siRNA per well using the Lipofectamine RNA iMAX Reagent (Invitrogen, Carlsbad, CA, USA).

For flow experiments, confluent monolayers of MEFs were seeded onto glass slides, and a parallel plate flow system was used to launch oscillatory flow ($0.5\pm4$ dyn/cm$^2$). The flow system was enclosed in a chamber (*Frangos et al., 1985*; *Fu et al., 2011*).

## Cell Lines

HEK293T (Cat NO. CRL-11268, ATCC) cells were cultured in DMEM medium with 10% FBS, penicillin (100 U/mL), and streptomycin (100 µg/mL). Cells were incubated at 37 °C in a humidified environment containing 5% CO$_2$ and grown to 70–80% confluence prior to treatment. Their identity has been authenticated by the supplier. HEK293T cells used here were negative for mycoplasma contamination test.

## Adenovirus and lentivirus production and infection

*Aff3ir-ORF2* sequences were inserted into the GV138 vector (CMV-MCS-3FLAG) to generate recombinant adenovirus (Ad-*Aff3ir-ORF2*). The short hairpin RNA (shRNA) sequences targeting mouse *Irf5* were 5'-GGGACAACACCATCTTCAAGG-3', 5'GGTTGCTGCTGGAGATGTTCT-3', and 5'-GCCTAGAGCAGTTTCTCAATG-3'. The control shRNA was 5'-GCGTGATCTTCACCGACAAGA-3'. These shRNAs were constructed and cloned into pLV-U6-shRNA-CMV-EGFP to generate recombinant lentivirus (lenti-shRNA- *Irf5* or lenti-shCtrl). MEFs were infected with adenovirus or lentivirus at a multiplicity of infection (MOI) of 10, with no detectable cellular toxicity observed.

## Western blot analysis

Whole-cell lysates were prepared in a lysis buffer containing a complete protease inhibitor cocktail, PhosSTOP, and PMSF (Roche, Mannheim, Germany). Cytoplasmic and nuclear proteins were extracted from wild-type and *Aff3ir-ORF2*$^{-/-}$ MEFs using a protein extraction kits (Invent Biotechnologies, SC-003, Beijing, China). Protein were separated by SDS-PAGE and transferred to nitrocellulose membranes (Cat. No. 10600001; GE Healthcare; Chicago, IL, USA). The membranes were incubated with primary antibodies. *Irf5* (Cat. No. 96527), *Irf8* (Cat. No. 98344), and Flag (Cat. No. 14793) antibodies were from Cell Signaling Technology (Danvers, MA, USA). *Icam1* (Cat. No. ab222736) antibodies were from Abcam (Cambridge, UK). *Aff3ir-ORF2* (Cat. No. C0302HL300-4) and *Aff3ir-ORF1* (Cat. No. C0302HL300) antibodies were from GenScript (Piscataway, NJ, USA). *Gapdh* (Cat. No. 60004–1-Ig) antibody was from Proteintech (Wuhan, China). *Aff3* (Cat. No. PA5-68961) antibody was from Thermo Fisher Scientific (Waltham, MA, USA).

After incubation with horseradish peroxidase-conjugated secondary antibodies, the proteins were visualized using enhanced chemiluminescence reagents in a ChemiScope3600 Mini chemiluminescence imaging system (Clinx Science Instruments; Shanghai, China). Protein levels were quantified by measuring integrated density with NIH Image J software (https://imagej.nih.gov/ij/), using *Gapdh* as a loading control for normalization.

## Co-immunoprecipitation

Whole-cell lysates were prepared by lysing cells in a 1% NP-40 lysis buffer containing 50 mM Tris-HCl, 1% Nonidet-P40, 0.1% SDS, and 150 mM NaCl, supplemented with a complete protease inhibitor cocktail (Cat. No. 04693132001; Roche, Indianapolis, IN, USA), a phosphatase inhibitor (PhosSTOP; Cat. No. 04906845001; Roche), and PMSF (Cat. No. IP0280; Solarbio Life Sciences; Beijing, China). Samples were incubated on ice for 30 min, then centrifuged at 12,000 g for 10 min, and the supernatant was transferred to a new tube. Protein concentrations were determined using the BCA Protein Assay Kit (Thermo Fisher Scientific, Grand Island, NY, USA).

For immunoprecipitation, 1000 µg of protein was incubated with specific antibodies at 4 °C for 12 hr with constant rotation. Subsequently, 50 µL of 50% Protein A/G PLUS-Agarose beads was added, and the incubation continued for an additional 2 hr. Beads were washed five times with the lysis buffer and collected by centrifugation at 12,000 g for 2 min at 4 °C. After the final wash, the supernatant was removed and discarded. Precipitated proteins were eluted by resuspending the beads in 2x

SDS PAGE loading buffer and boiling for 5 min. The eluates from immunoprecipitation were subjected to Western blot analysis.

## ELISA

The concentrations of *Il6* (EM0121) and *Il1b* (EM0109) in cell culture supernatant were measured using ELISA kit (FineTest, Wuhan, China). The experiments were conducted according to the protocols provided by the manufacturer.

## Total RNA extraction and real-time quantitative PCR analysis

Total RNA was extracted from cells using RNA extraction kits (Transgen Biotech, ER501-01, Beijing, China). Reverse transcription was performed with a reverse transcription kit (Thermo Fisher Scientific, Grand Island, NY, USA). Quantitative PCR was conducted using SYBR Select (Thermo Fisher Scientific) according to the manufacturer's protocol, with *Gapdh* serving as the internal control. The primers for quantitative real-time PCR are listed in *Supplementary file 2*.

## Luciferase reporter assay

The *Irf5*-binding motif and the full-length ZNF217 promoter were ligated into pGl3-based plasmids (Genechem, Shanghai, China), as previously described (*Qiao et al., 2022*). HEK293T cells were seeded into 24-well plates and grown to 70–80% confluency. Cells were transfected with the firefly luciferase reporter plasmid containing the *Irf5*-responsive ZNF217 promoter along with a β-galactosidase reporter plasmid (Promega, Madison, WI, USA) for 24 hr. Subsequently, cells were then infected with adenovirus (Ad-ORF2 or Ad-Scramble) for an additional 24 hr. Relative luciferase activity was measured using a luciferase assay and normalized to β-galactosidase activity as determined by the β-Galactosidase Enzyme Assay System (Promega, Madison, WI, USA).

## RNA-sequencing (RNA-seq)

RNA-seq was performed as we previously described (*Li et al., 2024*). Wild-type (WT) and *Aff3ir-ORF2*[-/-] MEFs were harvested, and RNA was extracted using the MagicPure Total RNA Kit (TransGen, Beijing, China). Whole transcriptome RNA-seq analysis were conducted by the Beijing Genomics Institute (BGI). Paired-end sequencing in 150 bp length was performed using the DNBSEQ-G400 platform. Raw data was filtered using SOAPnuke (v1.5.6). Differential gene expression analysis, with thresholds set at $p < 0.05$ and fold change ≥1.5, was performed via the BGI website (http://omiscribe.bgi.com). Pathway enrichment analysis was carried out using DAVID tools.

## Statistical analysis

Statistics analyses were performed using GraphPad Prism 8.0. No sample outliers were excluded. At least six independent experiments were performed for all biochemical experiments and the representative images were shown. Unpaired Student's *t*-test (two-tailed), one-way ANOVA, or two-way ANOVA with Bonferroni multiple comparison post hoc test were used for analyses, as appropriate. Sample size, statistical method, and statistical significance are specified in Figures and Figure Legends. Levels of probabilities less than 0.05 were regarded as significant.

# Acknowledgements

This work was supported by National Natural Science Foundation of China Grants (82330012, 82127808, 82422006, 32471166, and 82270516), British Heart Foundation (FS-15/74/31669), Natural Science Foundation of Tianjin (24JCJQJC00060 and 21JCYBJC01590), and the Key Research Project in Traditional Chinese Medicine of Tianjin (2023013).

# Additional information

## Funding

| Funder | Grant reference number | Author |
| --- | --- | --- |
| National Natural Science Foundation of China | 82330012 and 82127808 | Yi Zhu |
| National Natural Science Foundation of China | 82422006 | Jinlong He |
| National Natural Science Foundation of China | 32471166 | Jinlong He |
| National Natural Science Foundation of China | 82270516 | Jinlong He |
| British Heart Foundation | FS-15/74/31669 | Lingfang Zeng |
| Natural Science Foundation of Tianjin Municipality | 24JCJQJC00060 | Jinlong He |
| Natural Science Foundation of Tianjin Municipality | 21JCYBJC01590 | Lei Huang |
| Key Research Project in Traditional Chinese Medicine of Tianjin | 2023013 | Lei Huang |

The funders had no role in study design, data collection and interpretation, or the decision to submit the work for publication.

## Author contributions

Shuo He, Methodology, Writing – original draft, Writing – review and editing; Lei Huang, Software, Methodology; Zhuozheng Chen, Ze Yuan, Validation, Methodology; Yue Zhao, Formal analysis, Methodology; Lingfang Zeng, Conceptualization, Resources, Formal analysis; Yi Zhu, Conceptualization, Resources, Data curation, Formal analysis, Supervision, Project administration; Jinlong He, Conceptualization, Resources, Data curation, Formal analysis, Validation, Investigation, Project administration

## Author ORCIDs

Shuo He https://orcid.org/0009-0004-5912-9930
Lei Huang https://orcid.org/0000-0002-3911-0915
Yue Zhao https://orcid.org/0000-0002-6818-2352
Lingfang Zeng https://orcid.org/0000-0002-0390-4561
Jinlong He https://orcid.org/0000-0001-7349-8135

## Ethics

This study adhered to the Guide for the Care and Use of Laboratory Animals of the US National Institutes of Health (NIH Publication No. 85-23, revised 2011). All study protocols were approved by the Institutional Animal Care and Use Committee of Tianjin Medical University.

Reviewer #1 (Public review): https://doi.org/10.7554/eLife.103413.3.sa1
Reviewer #2 (Public review): https://doi.org/10.7554/eLife.103413.3.sa2
Author response https://doi.org/10.7554/eLife.103413.3.sa3

# Additional files

## Supplementary files

MDAR checklist

Supplementary file 1. The sequences of siRNAs.

Supplementary file 2. Primers for qRT-PCR.

Supplementary file 3. Expression of all differentially expressed genes from RNA-seq.

## Data availability

Sequencing data have been deposited in GEO under accession codes GSE286206. All data generated or analysed during this study are included in the manuscript and supporting files; source data files have been provided for all figures.

The following dataset was generated:

| Author(s) | Year | Dataset title | Dataset URL | Database and Identifier |
|---|---|---|---|---|
| He S, Zhu Y, He J | 2025 | Disruption of the Novel Nested Gene Aff3ir Mediates Disturbed Flow-Induced Atherosclerosis in Mice | https://www.ncbi.nlm.nih.gov/geo/query/acc.cgi?acc=GSE286206 | NCBI Gene Expression Omnibus, GSE286206 |

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
