## [Editor Report · eLife Assessment]

The study presents **valuable** findings on the role of Aff3ir, a gene implicated in flow-induced atherosclerosis and regulating the inflammation-associated transcription factor, IRF5. The in vivo data are **solid** in providing evidence on the role of Aff3ir in shear stress and formation of atheromatous plaques. The work will be of interest to clinical researchers and biologists focusing on inflammation and atherosclerosis in cardiovascular disease with a broad eLife readership.

---

## [Referee Report · Reviewer #1 (Public review)]

Summary:

The authors report the role of a novel gene Aff3ir-ORF2 in flow induced atherosclerosis. They show that the gene is anti-inflammatory in nature. It inhibits the IRF5 mediated athero-progression by inhibiting the causal factor (IRF5). Furthermore, authors show a significant connection between shear stress and Aff3ir-ORF2 and its connection to IRF5 mediated athero-progression in different established mice models which further validates the ex vivo findings.

Strengths:

(1) Adequate number of replicates were used for this study.

(2) Both in vitro and in vivo validation was done.

(3) Figures are well presented

(4) In vivo causality is checked with cleverly designed experiments

Weaknesses:

(1) Inflammatory proteins must be measured with standard methods e.g ELISA as mRNA level and protein level does not always correlate.

(2) RNA seq analysis has to be done very carefully. How does the euclidean distance correlate with the differential expression of genes. Do they represent neighborhood? If they do how does this correlation affect the conclusion of the paper?

(3) Volcano plot does not indicate q value of the shown genes. It is advisable to calculate q value for each of the genes which represents the FDR probability of the identified genes.

(4) GO enrichment was done against Global gene set or local geneset? Authors should provide more detailed information about the analysis.

(5) If the analysis was performed against global gene set. How does that connect with this specific atherosclerotic microenvironment?

(6) what was the basal expression of genes and how does the DGE (differential gene expression) values differ?

(7) How did IRF5 picked from GO analysis? was it within 20 most significant genes?

(8) Microscopic studies should be done more carefully? There seems to be a global expression present on the vascular wall for Aff3ir-ORF2 and the expression seems to be similar like AFF3 in fig 1.

Comments on Revision:

The authors have adequately addressed my concerns.

---

## [Referee Report · Reviewer #2 (Public review)]

Summary:

The authors recently uncovered a novel nested gene, Aff3ir, and this work sets out to study its function in endothelial cells further. Based on differences in expression correlating with areas of altered shear stress, they investigate a role for the isoform Aff3ir-ORF2 in endothelial activation and development of atherosclerosis downstream of disturbed shear stress. Using a knockout mouse model and in vivo overexpression experiments, they demonstrate a strong potential for Aff3ir-ORF2 to alleviate atherosclerosis. They find that Aff3ir-ORF2 interacts with the pro-inflammatory transcription factor IRF5 and retains it in the cytoplasm, hence preventing upregulation of inflammation-associated genes. The data expands our knowledge of IRF5 regulation which could be relevant to researchers studying various inflammatory diseases as well as adding to our understand of atherosclerosis development.

Strengths:

The in vivo data is convincing using immunofluorescence staining to assess AFF3ir-ORF2 expression, a knockout mouse model, overexpression and knockdown studies and rescue experiments in combination with two atherosclerotic models to demonstrate that Aff3ir-ORF2 can lessen atherosclerotic plaque formation in ApoE-/- mice.

Weaknesses:

The effect on atherosclerosis is clear and there is sufficient evidence to conclude that this is the result of reduced endothelial cell activation. However, other cell types such as smooth muscle cells or macrophages could be contributing to the effects observed. The mouse model is a global knockout and the shRNA knockdowns (Fig. 5) and overexpression data in Figure 2 are not cell type-specific. Only the overexpression construct in Figure 6 uses an ICAM-2 promoter construct, which drives expression in endothelial cells, though leaky expression of this promoter has been reported in the literature.

The in vitro experiments are solidly executed, but most experiments are performed in mouse embryonic fibroblasts (MEFs) and results extrapolated to endothelial cell responses. However, several key experiments are repeated in HUVEC, thereby making a solid case that Aff3ir-ORF2 can regulate IRF5 in both MEFs and HUVEC. It is important to note that the sequence of AFF3ir-ORF2 is not conserved in humans and lacks an initiation codon, hence the regulatory pathway is not conserved. However, the overexpression studies in HUVEC suggest that mouse AFF3ir-ORF2 can also regulate human IRF5 and hence the mechanism retains relevance for possible human health interventions.

Overall, the paper succeeds in demonstrating a link between Aff3ir-ORF2 and atherosclerosis. The study shows a functional interaction between Aff3ir-ORF2 and IRF5 in embryonic fibroblasts, but makes a solid case that this mechanism is relevant for atherosclerosis development via endothelial cell activation.

---

## [Author Response]

The following is the authors’ response to the original reviews

**Public Reviews:**

**Reviewer #1 (Public review):**
Summary:The authors report the role of a novel gene Aff3ir-ORF2 in flow-induced atherosclerosis. They show that the gene is anti-inflammatory in nature. It inhibits the IRF5-mediated athero-progression by inhibiting the causal factor (IRF5). Furthermore, the authors show a significant connection between shear stress and Aff3ir-ORF2 and its connection to IRF5 mediated athero-progression in different established mice models which further validates the ex vivo findings.Strengths:(1) An adequate number of replicates were used for this study.(2) Both in vitro and in vivo validation was done.(3) The figures are well presented.(4) In vivo causality is checked with cleverly designed experiments.

We thank you for your positive remarks.

Weaknesses:(1) Inflammatory proteins must be measured with standard methods e.g ELISA as mRNA level and protein level does not always correlate.

Thanks. We have followed your advice and performed ELISA experiments to measure the concentrations of inflammatory cytokines, including IL-6 and IL-1β. The newly acquired results have been included in Figure 2E (Line 160-163) in the revised manuscript.

(2) RNA seq analysis has to be done very carefully. How does the euclidean distance correlate with the differential expression of genes. Do they represent the neighborhood?

If they do how does this correlation affect the conclusion of the paper?

We thank the reviewer for this professional comments and apologize for the confusion. The heatmap using Euclidean distance was generated based on the expression levels of all differentially expressed genes (calculated with deseq2). Since its interpretation overlaps with the volcano plot presented in Figure 4B, we have moved the heatmap to Figure S5A in the revised manuscript and provided a detailed description in the figure legend (Lines 106-108 in the supporting information). Additionally, to better illustrate the variation among all samples, we have performed PCA analysis and included the new results in Figure 4A of the revised manuscript.

(3) The volcano plot does not indicate the q value of the shown genes. It is advisable to calculate the q value for each of the genes which represents the FDR probability of the identified genes.

Thank you for your careful review. We apologize for the incorrect labeling.

It was *P*.adj value. The label for Figure 4B has been corrected in the revised manuscript.

(4) GO enrichment was done against the Global gene set or a local geneset? The authors should provide more detailed information about the analysis.

Thank you. We performed GO enrichment analysis against the global gene set. The description of the results has been updated in the revised manuscript (Lines 222–224).

(5) If the analysis was performed against a global gene set. How does that connect with this specific atherosclerotic microenvironment?

Thank you for your insightful comments. We have followed your advice and investigated the functional characteristics of these differentially expressed genes in the context of the atherosclerotic microenvironment. The RNA-seq differential gene list was further mapped onto the atherosclerosis-related gene dataset (PMID: 27374120), resulting in 363 overlapping genes. The 363 genes were subjected to bioinformatics enrichment analysis using Gene Ontology (GO) databases. GO analysis of these genes revealed enrichment in processes related to cell−cell adhesion and leukocyte activation involved in immune response (Figure S5B), which is highly consistent with the observed effects of AFF3ir-ORF2 on VCAM-1 expression. The newly acquired data are presented in Figure S5B and the description of the results is included in the revised manuscript (Line 227-233).

(6) What was the basal expression of genes and how did the DGE (differential gene expression) values differ?

Thanks for the comments. The RNA-sequencing data has been submitted to GEO datasets (GSE286206), making the basal gene expression data available to readers.

The differential expression analysis was performed using DESeq2 (v1.4.5) (PMID: 25516281) with a criterion of 1.5-fold change and P<0.05. We has included the description in the revised manuscript in Lines 220-222 and Lines 575-576.

(7) How was IRF5 picked from GO analysis? was it within the 20 most significant genes?

Sorry for the confusion. IRF5 was not identified through GO analysis. To determine the upstream transcriptional regulators, we used the ChEA3 database to predict potential upstream transcription factors based on all differentially expressed genes. The top 20 transcription factors were selected based on their scores. To further explore their relationship with atherosclerosis, these top 20 transcription factors were mapped to the atherosclerosis-related gene list in the DisGeNET database. IRF5 and IRF8 were the only two overlapping genes. To clarify this process, we have included a more detailed description of the IRF prediction approach in the revised manuscript (Lines 234–239).

(8) Microscopic studies should be done more carefully? There seems to be a global expression present on the vascular wall for Aff3ir-ORF2 and the expression seems to be similar to AFF3 in Figure 1.

We thank the reviewer for the valuable suggestion. We have followed your advice and provided the more representative images in Figure 1F.

**Reviewer #2 (Public review):**
Summary:The authors recently uncovered a novel nested gene, Aff3ir, and this work sets out to study its function in endothelial cells further. Based on differences in expression correlating with areas of altered shear stress, they investigate a role for the isoform Aff3ir-ORF2 in endothelial activation and development of atherosclerosis downstream of disturbed shear stress. Using a knockout mouse model and in vivo overexpression experiments, they demonstrate a strong potential for Aff3ir-ORF2 to alleviate atherosclerosis. They find that Aff3ir-ORF2 interacts with the pro-inflammatory transcription factor IRF5 and retains it in the cytoplasm, hence preventing upregulation of inflammation-associated genes. The data expands our knowledge of IRF5 regulation which could be relevant to researchers studying various inflammatory diseases as well as adding to our understanding of atherosclerosis development.Strengths:The in vivo data is solid using immunofluorescence staining to assess AFF3ir-ORF2 expression, a knockout mouse model, overexpression and knockdown studies, and rescue experiments in combination with two atherosclerotic models to demonstrate that Aff3ir-ORF2 can lessen atherosclerotic plaque formation in ApoE^-/-^ mice.

We thank you for your positive remarks.

Weaknesses:While the in vivo data is generally convincing, a few data panels have issues and will need addressing. Also, the knockout mouse model will need to be described, since the paper referred to in the manuscript does not actually report any knockout mouse model. Hence it is unclear how Aff3ir-ORF2 is targeted, but Figure S2B shows that targeting is partial, since about 30% expression remains at the RNA level in MEFs isolated from the knockout mice.

We thank you for the valuable comments.

First, we have followed your advice and included detailed information regarding the animal construction in the revised manuscript in Line 405-415. Additionally, the genotyping results have been included in new Figure S3A.

Second, we acknowledge your concern about the knockout efficiency of ORF2 in mice. While the PCR assay indicated approximately 30% residual expression, our Western blot analysis of aorta samples demonstrated that ORF2 protein was barely detectable in knockout mice, as shown in new Figure S3B-C. Besides, our in vivo experiments using MEF from WT and AFF3ir-ORF2^-/-^ mice (Figure 4I) further confirmed successful knockout.

Third, we have included a discussion addressing the discrepancies between PCR and Western blot results. In addition to technical differences between the two methods, the nature of AFF3ir-ORF2 may also contribute to these inconsistencies. The parent gene AFF3 is located in a genetically variable region and can be excised via intron 5 to form a replicable transposon, which translocates to other chromosomes and has been linked to leukemia (PMID: 34995897, 12203795, 12743608, and 17968322). AFF3ir is located in the intron 6, thus it exists in the transposon, which may complicate the measurement of its expression. Replicable transposons can exist as extrachromosomal elements, allowing them to be inherited across generations. We have included these discussion in the revised manuscript in Line 188-196.

While the effect on atherosclerosis is clear, the conclusion that this is the result of reduced endothelial cell activation is not supported by the data. The mouse model is described as a global knockout and the shRNA knockdowns (Figure 5) and overexpression data in Figure 2 are not cell type-specific. Only the overexpression construct in Figure 6 uses an ICAM-2 promoter construct, which drives expression in endothelial cells, though leaky expression of this promoter has been reported in the literature. Therefore, other cell types such as smooth muscle cells or macrophages could be responsible for the effects observed.

Thank you for your critical comment. To address your concern, we have made the following three revisions:

First, we have analyzed the expression of AFF3ir-ORF2 in the vascular wall with or without intima in WT and AFF3ir-ORF2 knockout mice. As shown in Figure 1B and Figure S1A, while the expression of AFF3ir-ORF2 was notably downregulated in the aortic intima of athero-prone regions compared to the protective region, it remained largely unchanged in the aortic wall without intima across different regions of the aorta. This suggested that AFF3ir-ORF2 might play a predominant role in endothelial cells rather than other cell types in the context of shear stress.

Second, we have used human endothelial cells (HUVECs) to further confirm our findings. As shown in Figure 2C and Figure S2B, we found that AFF3ir-ORF2 overexpression could attenuate disturbed shear stress-induced IRF5 nuclear translocation and the expression of inflammatory genes in HUVECs, suggesting the potential anti-inflammatory effects of AFF3ir-ORF2 in endothelial cells.

Third, we agree with the reviewer’s comment that we cannot completely exclude the potential involvement of other cell types. Hence, we have included a limitation statement in the discussion part in Lines 341-344.

The weakest part of the manuscript is the in vitro experiment using some nonidentifiable expression differences. The data is used to hypothesise on a role for IRF5 in the effects observed with Aff3ir-ORF2 knockout.

Thank you for the comments. To address your concerns, we have made the following two changes:

First, we have further investigated the functional features of the differential genes from the RNA-seq in the context of atherosclerotic microenvironment. The differential gene list was mapped onto the atherosclerosis-related gene dataset (PMID: 27374120), and a total of 363 genes overlapped. These 363 genes were subjected to bioinformatics enrichment analysis using Gene Ontology (GO) databases. GO analysis showed that these genes were mainly enriched in cell−cell adhesion and leukocyte activation involved in immune response, which aligns with the expression of VCAM-1 affected by AFF3ir-ORF2. The newly acquired data are presented in Figure S5B and the description of the results has been updated in the revised manuscript (Line 227-233).

Second, we have further verified the RNA-seq results in vitro. Several classical inflammatory factors, including ICAM-1, CCL5, and CXCL10, which mRNA levels were significantly downregulated in RNA-seq and were also identified as target genes of IRF5, were analyzed. We found that AFF3ir-ORF2 deficiency aggravated, while AFF3ir-ORF2 overexpression attenuated, the expression of ICAM-1, CCL5, and CXCL10 induced by disturbed shear stress (New Figure S5D). Besides, the regulation of ICAM-1 by AFF3ir-ORF2 was confirmed at both protein and mRNA levels in HUVECs (Figure 2C-D and Figure S2B).

Overall, the paper succeeds in demonstrating a link between Aff3ir-ORF2 and atherosclerosis, but the cell types involved and mechanisms remain unclear. The study also shows a functional interaction between Aff3ir-ORF2 and IRF5 in embryonic fibroblasts, but any relevance of this mechanism for atherosclerosis or any cell types involved in the development of this disease remains largely speculative.

Thank you for all the valuable comments. The specific responses have been provided above. Briefly, we have followed your advice and further confirmed the regulation of AFF3ir-ORF2 on IRF5 in endothelial cells. Besides, the RNA-seq results have been further analyzed, and partial results have been verified in endothelial cells to support the anti-inflammatory role of AFF3ir-ORF2. We greatly appreciate the reviewer’s insightful comments, which guided our revisions and contributed to significantly improving the paper.

**Reviewer #3 (Public review):**
This study is to demonstrate the role of Aff3ir-ORF2 in the atheroprone flow-induced EC dysfunction and ensuing atherosclerosis in mouse models. Overall, the data quality and comprehensiveness are convincing. In silico, in vitro, and in vivo experiments and several atherosclerosis were well executed. To strengthen further, the authors can address human EC relevance.

We thank you for your positive remarks and insightful comments.

Major comments:(1) The tissue source in Figures 1A and 1B should be clarified, the whole aortic segments or intima? If aortic segment was used, the authors should repeat the experiments using intima, due to the focus of the current study on the endothelium.

We thank you for the suggestion. The tissue used in Figures 1A and 1B was from aortic intima. The description has been updated for clarity in the revised manuscript on Lines 114-125.

(2) Why were MEFs used exclusively in the in vitro experiments? Can the authors repeat some of the critical experiments in mouse or human ECs?

Thank you for this insightful comment. Isolation and culture of mouse primary aortic ECs were notorious technically difficult and shear stress experiment require a large number of cells. Considering MEFs exhibit responses consistent with those of ECs, which has been delicately proved (PMID: 23754392), we used MEFs in our in vitro experiments.

However, following your valuable advice, we have now employed human ECs (HUVECs) to confirm our findings. Consistent with our results in MEFs, we found that AFF3ir-ORF2 overexpression reduced the expression of inflammatory genes induced by disturbed shear stress at both protein and mRNA levels in HUVECs (Figure 2C, Figure S2B). Notably, despite the significant anti-inflammatory effects of AFF3irORF2, the sequence of this gene is not conserved in *Homo sapiens* and lacks an initiation codon, which is why we did not further proceed with the loss-of-function experiments.

(3) The authors should explain why AFF3ir-ORF2 overexpression did not affect the basal level expression of ICAM-1, VCAM-1, IL-1b, and IL-6 under ST conditions (Figure 2A-C).

We thank you for raising this critical question. Indeed, we found that AFF3ir-ORF2 overexpression did not affect the basal level of inflammatory genes under ST conditions, while it exerted anti-inflammatory effects under OSS conditions. One underlying reason might be the relative low level of expression of inflammatory genes under ST compared to OSS conditions. Additionally, as our findings suggested, AFF3ir-ORF2 exerted its anti-inflammatory role by binding to IRF5 and inhibiting IRF5 nuclear translocation. However, as shown in Figure 4I, IRF5 might be predominantly localized in the cytoplasm rather than the nucleus under ST conditions.

We have included the description in the revised manuscript on Lines 157-163.

(4) Please include data from sham controls, i.e., right carotid artery in Figure 2E.

Thank you for the suggestion. We have followed your advice and included sham controls (staining of the right carotid arteries) in Figure S2E.

(5) Given that the merit of the study lies in the effect of different flow patterns, the legion areas in AA and TA (Figure 3B, 3C) should be separately compared.

We have followed your valuable suggestion and included the additional statistical results in Figure 3C in the revised manuscript.

(6) For confirmatory purposes for the variations of IRF5 and IRF8, can the authors mine available RNA-seq or even scRNA-seq data on human or mouse atherosclerosis? This approach is important and could complement the current results that are lacking EC data.

Thank you for your valuable suggestion. In the present study, we found that disturbed flow did not alter the protein level of IRF5 but promoted its nuclear translocation. Following your advice, we analyzed the expression of IRF5 in human ECs (GSE276195) and atherosclerotic mouse arteries (GSE222583) using public databases. Consistently, IRF5 did not show significant changes in mRNA levels under these conditions (Figure S5E-F), suggesting that the regulation of IRF5 in the context of disturbed flow or atherosclerosis is primarily post-translational.

(7) With the efficacy of using AAV-ICAM2-AFF3ir-ORF2 in atherosclerosis reduction (Figure 6), the authors are encouraged to use lung ECs isolated from the AFF3ir-ORF2/-mice to recapitulate its regulation of IRF5.

We greatly appreciate your valuable suggestion to use lung ECs from mice. We have observed that AFF3ir-ORF2 deficiency enhanced the nuclear translocation of IRF5 induced by OSS. Noteworthy, the transcriptional levels of IRF5 were minimally affected by AFF3ir-ORF2 deficiency. Hence, to recapitulate the regulation of IRF5 with lung ECs isolated from the AFF3ir-ORF2^-/-^ mice, it would require treating lung ECs with OSS followed by isolation of subcellular components. However, both in vitro shear stress treatment and subcellular fraction isolation require a large number of cells, and mouse lung ECs are difficult to culture and pass through several passages. Therefore, we hope the reviewer understands that these experiments were not performed. As an alternative, we have confirmed the transcriptional activity changes of IRF5 due to AFF3ir-ORF2 manipulation by analyzing the expression of its target genes indicated from RNA-seq results in both the intima of mouse aorta (Figure S5C-D) and HUVECs (Figure 2C-D and Figure S2B). Our findings show that AFF3ir-ORF2 deficiency increases, while its overexpression decreases, the expression levels of IRF5-targeted genes in endothelial cells.

**Recommendations for the authors:**

**Reviewer #1 (Recommendations for the authors):**
Figure 2H - As I understand it, this is MFI measurement of VCAM. Please change accordingly.

Thanks. Corrected.

**Reviewer #2 (Recommendations for the authors):**
My major concern is the use of MEFs for all in vitro experiments. All experiments should be done in endothelial cells if the aim is to show a mechanism relevant to endothelial activation and atherosclerosis. Lines 314-316 of the conclusion are absolutely not supported by the data.

Thank you for the insightful comment. Following your advice, we have employed human ECs (HUVECs) to confirm our findings. Consistent with the findings in MEFs, we found that AFF3ir-ORF2 decreased the expression of inflammatory genes induced by disturbed shear stress, both at protein and mRNA levels in HUVECs (Figure 2C, Figure S2B).

Since the in vivo experiments are not cell type-specific, it would be important to test and compare the expression of Aff3ir-ORF2 in endothelial cells as well as smooth muscle and macrophages to support any claim of cell type involvement in the effects observed.

We thank you for the valuable suggestion. In the revised manuscript, we have followed your suggestion and analyzed the expression pattern of AFF3ir-ORF2 in different regions of the aorta with or without endothelium. We observed a marked reduction in AFF3ir-ORF2 expression in the intima of the aortic arch compared to that in the intima of the thoracic aorta (Figure 1B-C). In contrast, the expression of AFF3irORF2 in the media and adventitia was comparable between the aortic arch and thoracic aorta (Figure S1A-B). These findings provide further evidence supporting the predominant role of endothelial cells. The description has been modified accordingly in the revised manuscript on Lines 121-134.

The results of the RNA-seq experiment should be disclosed. The experiment should be deposited on GEO or similar and a table of differentially expressed genes added to the manuscript.

Thank you for the suggestion. We have followed your advice and submitted the RNA-sequencing data to GEO datasets (GSE286206). Besides, a table of differentially expressed genes has been included in the revised manuscript as Table S3.

Minor comments:(1) Figure 1A. Missing the labels of the target.

Thanks. Corrected.

(2) Figure 1D. Cell alignment in AA compared to TA suggests that the image is of the outer curvature, but Figure 1F is showing that the outer curvature is expressing more ORF2 than the inner. Why was the outer curvature chosen for this panel and is it true to conclude on that assumption that expression of ORF2 compares as TA > Outer > Inner curvature?

We thank you for the insightful suggestion. We have followed your advice and performed en-face immunofluorescence staining of AFF3ir-ORF2 and quantification of AFF3ir-ORF2 expression in AA inner, AA outer, and TA regions. As shown in new Figure 1D-E, the results indeed indicated that expression of AFF3irORF2 compares as TA > AA outer > AA inner.

(3) Figure 2H. Target mislabelled as ICAM-1 instead of VCAM-.

Thanks. Corrected.

(4) Figure S1A. VE-cad staining and cell shape differ between control and overexpression. Is this a phenotype or are different areas of the vasculature shown, which would make it hard to interpret since Aff3ir-ORF2 levels differ in different vessel areas?

We thank the reviewer for raising this important question. For Figure S1A, only common carotid arteries were used for the staining. The potential differences in cell shape observed might be due to variations in the procedure during immunofluorescence staining. To avoid any misinterpretation, more representative images have been provided in the revised Figure S2C.

(5) Figure 3D-G. Images are not representative of the quantification results.

Thank you. More representative images have been replaced in the revised Figure 3D and Figure 3F.

(6) Line 220. Data for IRF8 are not shown in the figure to support this claim.

Thank you for pointing this out. The expression level of IRF8 has been included in Figure S5C.

(7) Figure 6F. AAV-AFF3ir-ORF2 panel order inverted.

Thanks. Corrected.

(8) Line 401. Type "hat" instead of "h at".

Sorry for the typo. Corrected.

**Reviewer #3 (Recommendations for the authors):**
Minor comments:(1) The rationale for the following sentence (lines 126-128) is lacking: "Moreover, 126 we observed the expression of AFF3ir-ORF2 in longitudinal sections of the mouse aorta (B. 127 Li et al., 2019)".

Thanks. The rationale for these experiments have been included in the revised manuscript on Line 127-129.

(2) The source of antibodies against AFF3ir-ORF1 and AFF3ir-ORF2 used in western blot and immunostaining experiments were not mentioned in the manuscript.

Thanks. The antibody information has been included in the method part on Line 456-457, 510-511.

(3) The rationale and data interpretation is not clear for the following sentence (lines 220-221): "In addition, neither IRF5 nor IRF8 expression was regulated by AFF3irORF2 220 (Figure 4F)".

Thank you for pointing this out. The expression level of IRF8 has been included in Figure S5C. The sentence has been modified accordingly on Lines 253254.

(4) The quality of AFF3ir-ORF2 blot in Figure 4I needs improvement.

Thanks. More representative images have been included in Figure 4I.

(5) It appears that AFF3ir-ORF2 was present in both cytoplasm and nucleus. Does AFF3ir-ORF2 have a nuclear entry peptide? Also, the nuclear entry of AFF3ir-ORF2 can be enhanced by an immunofluorescence staining experiment.

Thank you for your insightful comments. Indeed, although we did not observe any significant subcellular changes in the localization of AFF3ir-ORF2 under shear stress conditions, our immunostaining results revealed that AFF3ir-ORF2 is localized in both the cytoplasm and nucleus. To explore whether AFF3ir-ORF2 contains nuclear localization signals, we utilized the NLStradamus tool (http://www.moseslab.csb.utoronto.ca/NLStradamus/) to analyze its sequence. The predication indicated that AFF3ir-ORF2 lacks a nuclear localization signal.